# CENP-C-Mis12 complex establishes a regulatory loop through Aurora B for chromosome segregation

Weixia Kong[1], Masatoshi Hara[1], Yurika Tokunaga[2], Kazuhiro Okumura[2], Yasuhiro Hirano[1], Jiahang Miao[1], Yusuke Takenoshita[1], Masakazu Hashimoto[1,3], Hiroshi Sasaki[1], Toshihiko Fujimori[4,5], Yuichi Wakabayashi[2], Tatsuo Fukagawa[1]

**Establishing the correct kinetochore–microtubule attachment is crucial for faithful chromosome segregation. The kinetochore has various regulatory mechanisms for establishing correct bipolar attachment. However, how the regulations are coupled is not fully understood. Here, we demonstrate a regulatory loop between the kinetochore protein CENP-C and Aurora B kinase, which is critical for the error correction of kinetochore–microtubule attachment. This regulatory loop is mediated through the binding of CENP-C to the outer kinetochore Mis12 complex (Mis12C). Although the Mis12C-binding region of CENP-C is dispensable for mouse development and proliferation in human RPE-1 cells, those cells lacking this region display increased mitotic defects. The CENP-C-Mis12C interaction facilitates the centromeric recruitment of Aurora B and the mitotic error correction in human cells. Given that Aurora B reinforces the CENP-C-Mis12C interaction, our findings reveal a positive regulatory loop between Aurora B recruitment and the CENP-C-Mis12C interaction, which ensures chromosome biorientation for accurate chromosome segregation.**

## Introduction

Accurate chromosome segregation during mitosis is crucial for the transmission of genomic information to the daughter cells. Failure of this process causes chromosomal instability, leading to aneuploidy, a hallmark of cancer (Bakhoum & Cantley, 2018; Ben-David & Amon, 2020; Vishwakarma & McManus, 2020; Girish et al, 2023; Baker et al, 2024).

The kinetochore assembled on the centromere is a large protein complex that establishes a bioriented chromosome–microtubule attachment for accurate chromosome segregation during mitosis and meiosis. The kinetochore is composed of two major complexes:

the constitutive centromere-associated network (CCAN) at the inner kinetochore and KNL1, Mis12, and Ndc80 complexes (KMN network, KMN for short) at the outer kinetochore (Fukagawa & Earnshaw, 2014; McKinley & Cheeseman, 2016; Hara & Fukagawa, 2017, 2018; Mellone & Fachinetti, 2021). The vertebrate CCAN, a 16-protein complex, associates with the centromeric chromatin containing the histone H3 variant CENP-A and establishes a base of the kinetochore structure (Palmer et al, 1987; Foltz et al, 2006; Izuta et al, 2006; Okada et al, 2006; Hori et al, 2008; Amano et al, 2009; Black & Cleveland, 2011; Nishino et al, 2012; Westhorpe & Straight, 2013). During the late G2 and M phases, KMN, the major microtubule-binding complex, is recruited onto CCAN. Consequently, the fully assembled mitotic kinetochore connects the chromosomes with spindle microtubules for chromosome segregation (Cheeseman et al, 2006; DeLuca et al, 2006; Alushin et al, 2010; McKinley & Cheeseman, 2016; Nagpal & Fukagawa, 2016; Pesenti et al, 2016; Hara & Fukagawa, 2017).

The CCAN protein CENP-C is a conserved essential protein for chromosome segregation (Earnshaw & Rothfield, 1985; Tomkiel et al, 1994; Meluh & Koshland, 1995; Fukagawa & Brown, 1997; Kalitsis et al, 1998; Fukagawa et al, 1999) and is thought to be a central hub of kinetochore assembly because CENP-C binds to proteins in all layers of the kinetochore, including CENP-A, other CCAN proteins, and the Mis12 complex (Mis12C) of KMN through its multiple functional domains (Screpanti et al, 2011; Kato et al, 2013; Klare et al, 2015). The interaction between CENP-C and Mis12C was reconstituted in vitro, and structural analyses identified key residues of CENP-C for Mis12-binding, which are conserved among species (Dimitrova et al, 2016; Petrovic et al, 2016). The CENP-C-Mis12C interaction is enhanced by Aurora B–mediated phosphorylation of DSN1, a component of Mis12C, and this phosphoregulation is conserved between yeasts and vertebrates (Kim & Yu, 2015; Rago et al, 2015; Dimitrova et al, 2016; Petrovic et al, 2016; Hara et al, 2018; Bonner et al, 2019). Mis12C also interacts with the microtubule-binding Ndc80 complex (Ndc80C) (Cheeseman et al, 2006; Kline et al, 2006). Therefore, the CENP-C-Mis12C interaction appears to be

[1]Graduate School of Frontier Biosciences, Osaka University, Osaka, Japan   [2]Division of Experimental Animal Research, Cancer Genome Center, Chiba Cancer Center Research Institute, Chiba, Japan   [3]Department of Cell Science, Institute of Biomedical Sciences, School of Medicine, Fukushima Medical University, Fukushima, Japan   [4]Division of Embryology, National Institute for Basic Biology, Okazaki, Japan   [5]Basic Biology Program, The Graduate University for Advanced Studies, Okazaki, Japan

Correspondence: m.hara.fbs@osaka-u.ac.jp; fukagawa.tatsuo.fbs@osaka-u.ac.jp

crucial for bridging centromeres with microtubules. However, this was not the case in chicken DT40 cells. We previously showed that deletion of the Mis12C-binding domain (M12BD) of CENP-C resulted in no apparent growth defects in chicken DT40 cells (Hara et al, 2018). Instead, deletion of the KMN-binding domain from CENP-T, another CCAN protein, causes severe mitotic arrest and consequent cell death, indicating that KMN binding of CENP-T is essential for bioriented chromosome–microtubule attachment in DT40 cells (Hara et al, 2018).

However, given that the key residues in the M12BD of CENP-C and Aurora B–mediated regulation are well conserved among species, unrevealed advantages to the CENP-C-Mis12C interaction that could not be detected in our previous systems may exist. Clarifying the benefits of the CENP-C-Mis12C interaction and elucidating its physiological role are crucial for understanding kinetochore regulation leading to faithful chromosome segregation.

To address these issues, we generated and characterized mice lacking the M12BD of CENP-C ($Cenp c^{\Delta M12BD}$). Although the M12BD was largely dispensable for mouse development, the $Cenp c^{\Delta M12BD/\Delta M12BD}$ mice were caner-prone in the two-stage skin carcinogenesis model, suggesting the contribution of the CENP-C-Mis12C interaction to cancer prevention. This is in line with increased mitotic defects in MEFs established from $Cenp c^{\Delta M12BD/\Delta M12BD}$ embryos. To further investigate its molecular mechanisms, we used human RPE-1 cells and found that deletion of the M12BD reduced the centromeric localization of Aurora B during mitosis. Given that Aurora B kinase regulates kinetochore–microtubule interactions, M12BD deletion resulted in impaired error correction of kinetochore–microtubule attachment, suggesting that the CENP-C-Mis12C interaction positively regulates mitotic Aurora B localization to establish chromosome biorientation. We further clarified the regulatory axis for Aurora B localization using HeLa cells, which are cancerous cells with chromosomal instability and low Aurora B kinase activity at the mitotic centromeres. Forced binding of Mis12C to CENP-C increased Aurora B levels at centromeres and improved error correction efficiency in HeLa cells. Given that Aurora B facilitates the CENP-C-Mis12C interaction (Kim & Yu, 2015; Rago et al, 2015; Dimitrova et al, 2016; Petrovic et al, 2016), we propose a positive regulatory loop between the CENP-C-Mis12C interaction and Aurora B recruitment at the centromeres to maintain Aurora B kinase activity for efficient mitotic error correction to establish bipolar attachment and subsequent faithful chromosome segregation during mitosis.

# Results

## Mis12C-binding domain of CENP-C is dispensable for mouse development but is required for proper mitotic progression in MEFs

CENP-C binds to Mis12C via its N-terminal region (Przewloka et al, 2011; Screpanti et al, 2011; Dimitrova et al, 2016; Petrovic et al, 2016; Hara et al, 2018) (Mis12C-binding domain: M12BD; Figs 1A and S1A). We previously found that the M12BD is dispensable for the proliferation of chicken DT40 cells (Hara et al, 2018). However, given the

amino acid (aa) sequence conservation of the M12BD among species and its importance in KMN interactions (Figs 1A and S1A), we wondered whether this finding was specific to DT40 cells.

To test whether CENP-C requires the M12BD for its functions in other species and to determine the physiological importance of the CENP-C-Mis12C interaction, we generated a mutant mouse model lacking the M12BD of CENP-C. Using the CRISPR/Cas9 system, we deleted exons 2–4 from the *Cenpc1* (*Cenpc*) gene, which encodes aa 7–75 (Fig S1B). Their deletion removed most of the M12BD, including the key conserved residues for Mis12C binding (CENP-C$^{\Delta M12BD}$; Fig S1A and B) (Screpanti et al, 2011; Petrovic et al, 2016). In contrast to *Cenpc* null mice, which do not survive embryonic development (Kalitsis et al, 1998), intercrossing heterozygous (*Cenpc$^{+/\Delta M12BD}$*) mice produced homozygous progeny (*Cenpc$^{\Delta M12BD/\Delta M12BD}$*), despite a slight reduction in female offspring (Figs 1B and S1C). These results suggest that the M12BD of CENP-C is largely dispensable in mouse development; however, female embryos are sensitive to the loss of the M12BD.

Chromosomal instability and subsequent micronucleus formation cause female-biased lethality in mouse embryos through anti-inflammatory activity of testosterone in male embryos (McNairn et al, 2019). This prompted us to investigate mitotic progression in MEFs established from E14.5 *Cenpc$^{\Delta M12BD/\Delta M12BD}$* embryos (Fig S1D and E). We also prepared MEFs from *Cenpc$^{+/+}$* and *Cenpc$^{+/\Delta M12BD}$* mice and confirmed their genotypes and CENP-C protein expression (Fig S1E).

We examined the growth of MEFs and found that the proliferation of *Cenpc$^{\Delta M12BD/\Delta M12BD}$* MEFs was slower than that of *Cenpc$^{+/+}$* or *Cenpc$^{+/\Delta M12BD}$* MEFs (Fig 1C). Next, we examined the mitotic progression of MEFs using time-lapse imaging (Fig 1D). It took ~16 min from the nuclear envelope breakdown to the anaphase onset in *Cenpc$^{+/+}$* and *Cenpc$^{+/\Delta M12BD}$* MEFs. The mitotic progression was slightly, but significantly, delayed in *Cenpc$^{\Delta M12BD/\Delta M12BD}$* MEFs (Fig 1D and E). We also observed an increase in the population with chromosome missegregation during anaphase (lagging chromosome or chromosome bridge) in *Cenpc$^{\Delta M12BD/\Delta M12BD}$* MEFs and in the population with micronuclei in *Cenpc$^{\Delta M12BD/\Delta M12BD}$* MEFs (Fig 1F and G). Consistent with these observations, the population with more than 4C DNA content increased in *Cenpc$^{\Delta M12BD/\Delta M12BD}$* MEFs (Fig 1H). These results showed that *Cenpc$^{\Delta M12BD/\Delta M12BD}$* MEFs exhibited chromosomal instability, suggesting that the M12BD of CENP-C contributes to accurate chromosome segregation in mouse cells despite being largely dispensable for development.

## Deletion of the M12BD of CENP-C accelerates tumor formation and malignant conversion in the two-stage skin carcinogenesis model

The chromosomal instability is a hallmark of cancer (Bakhoum & Cantley, 2018; Ben-David & Amon, 2020; Vishwakarma & McManus, 2020; Girish et al, 2023; Baker et al, 2024). Because the deletion of the M12BD from CENP-C led to increased chromosomal instability in MEFs, we examined the cancer susceptibility of *Cenpc$^{\Delta M12BD/\Delta M12BD}$* mice using a two-stage skin carcinogenesis model (Fig 2A) (Kemp, 2005; Abel et al, 2009). Between 14 and 20 wk after the initial 7,12-dimethylbenz(a)anthracene (DMBA)/12-O-tetradecanoylphorbol-13-acetate (TPA) treatment, *Cenpc$^{\Delta M12BD/\Delta M12BD}$* mice formed significantly more papillomas than *Cenpc$^{+/+}$* or *Cenpc$^{+/\Delta M12BD}$* mice

A

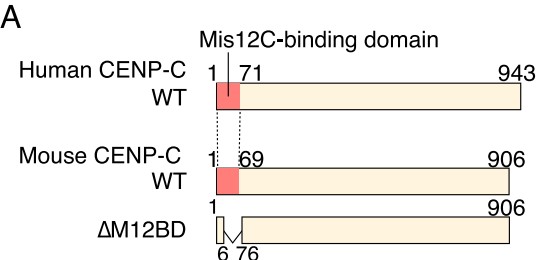

Mis12C-binding domain

Human CENP-C
WT
1 | 71 943

Mouse CENP-C
WT
1 | 69 906

ΔM12BD
1 906
6 76

B

| | $Cenpc^{+/+}$ | $Cenpc^{+/ΔM12BD}$ | $Cenpc^{ΔM12BD/ΔM12BD}$ | Total |
|---|---|---|---|---|
| Male | 45 | 64 | 30 | 139 |
| Female | 44 | 67 | 13*** | 124 |
| Total | 89 (34%) | 131 (50%) | 43 (16%)*** | 263 (100%) |

C

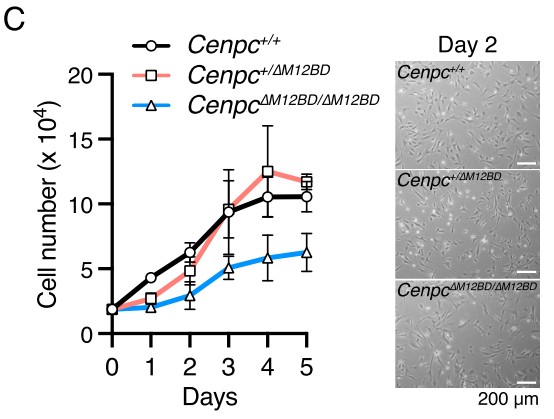

D

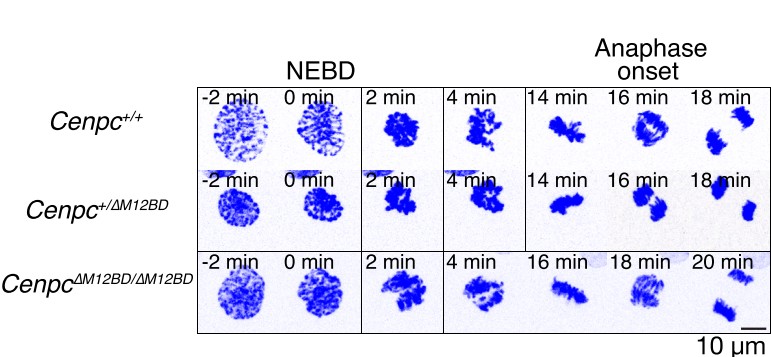

E

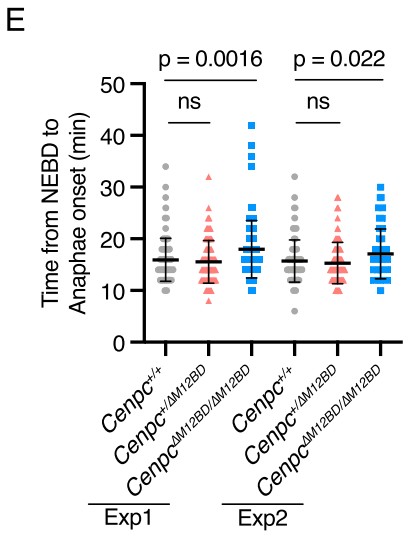

F

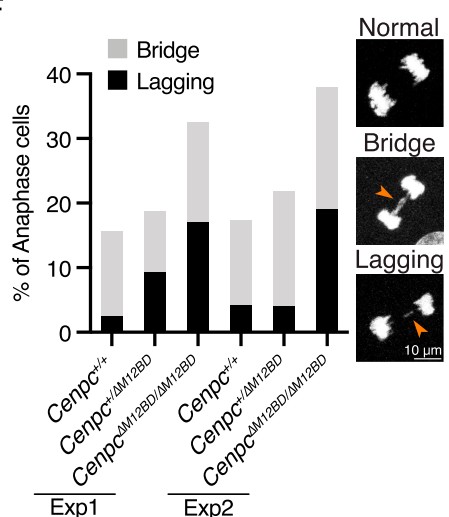

G

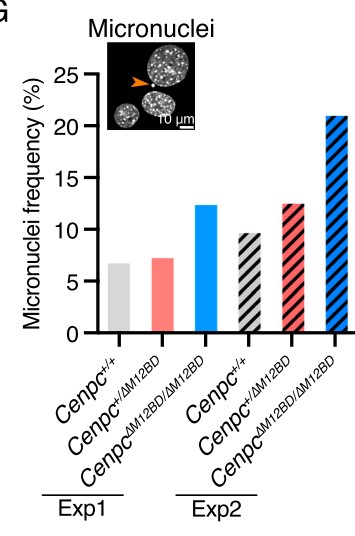

H

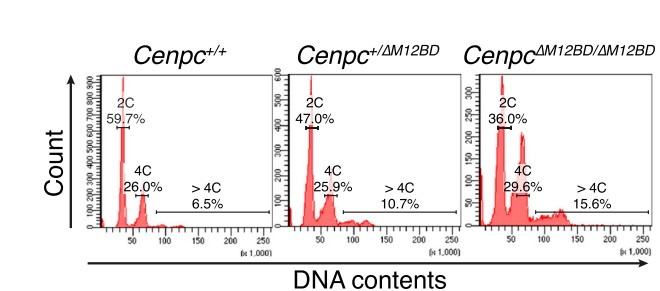

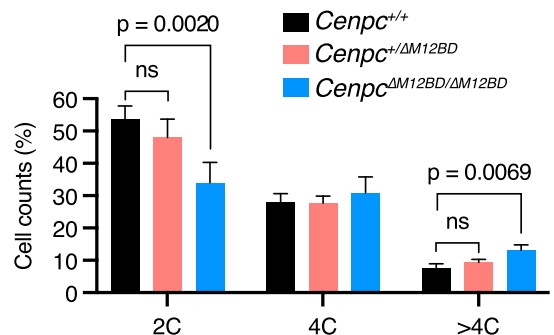

(Fig 2B and C). We also monitored mice after 20 wk and found that malignant conversion was significantly promoted in $Cenpc^{\Delta M12BD/\Delta M12BD}$ mice by 36 wk compared with that in $Cenpc^{+/+}$ or $Cenpc^{+/\Delta M12BD}$ mice (Fig 2D). These results demonstrate that $Cenpc^{\Delta M12BD/\Delta M12BD}$ mice are cancer-prone, suggesting that the M12BD of CENP-C contributes to cancer prevention.

## M12BD deletion from CENP-C causes chromosome missegregation, leading to mitotic defects in human RPE-1 cells

We aimed to understand the molecular mechanisms by which the CENP-C M12BD prevents chromosomal instability. Because MEFs are heterogeneous and their cell proliferation is sensitive to replicative senescence, which limits detailed analyses of mitotic regulation, we used human retinal epithelial cells (RPE-1) for further investigation. RPE-1 is a widely used noncancerous cell line with a stable near-diploid karyotype. We generated RPE-1 cells in which endogenous CENP-C was replaced with FLAG-human CENP-C lacking the Mis12C-binding domain (aa 1–75 region; M12BD; Fig 3A). These cells were referred to as CENP-C$^{\Delta M12BD}$ RPE-1 cells (Figs 3A and S2A–G). We also generated RPE-1 cells expressing WT FLAG-human CENP-C (CENP-C$^{WT}$ RPE-1 cells) as controls (Figs 3A and S2A–G).

First, we quantified the Mis12C levels in the kinetochores of CENP-C$^{\Delta M12BD}$ RPE-1 cells. For this, we immunostained DSN1, a component of Mis12C, in CENP-C$^{\Delta M12BD}$ or CENP-C$^{WT}$ RPE-1 cells expressing mScarlet-CENP-A as a kinetochore marker (Fig S2A). As shown in Fig 3B, the punctate DSN1 signals found in CENP-C$^{WT}$ cells were significantly reduced in CENP-C$^{\Delta M12BD}$ RPE-1 cells. We also examined the levels of KNL1 and Ndc80 complexes (KNL1C and Ndc80C) by immunostaining with antibodies against their components (KNL1 and Hec1, respectively) and found that the signals of both KNL1C and Ndc80C were reduced in CENP-C$^{\Delta M12BD}$ RPE-1 cells (Fig 3C and D). The reduction of Ndc80C levels was mild compared with that of Mis12C and KNL1C levels. This can be explained by three additional Ndc80C-binding sites in CENP-T (Huis In't Veld et al, 2016; Rago et al, 2015; Takenoshita et al, 2022).

Next, we examined whether CENP-C$^{\Delta M12BD}$ RPE-1 cells showed mitotic defects as observed in $Cenpc^{\Delta M12BD/\Delta M12BD}$ MEFs. In contrast to MEFs, in which the deletion of the M12BD from CENP-C delayed cell growth, CENP-C$^{\Delta M12BD}$ RPE-1 cells grew comparably to CENP-C$^{WT}$ cells (Fig S2H). However, time-lapse imaging showed that the mitotic progression from the nuclear envelope breakdown to the anaphase onset was significantly delayed in CENP-C$^{\Delta M12BD}$ RPE-1 cells, as observed in $Cenpc^{\Delta M12BD/\Delta M12BD}$ MEFs (Fig 3E and F). We also observed an increase in chromosome missegregation with lagging or bridging chromosomes in CENP-C$^{\Delta M12BD}$ RPE-1 cells (Fig 3G). In addition, the cell population with micronuclei was increased in CENP-C$^{\Delta M12BD}$ RPE-1 cells (Fig 3H).

## Reduced Ndc80C and kinetochore-associated microtubules (K-fiber) do not cause significant mitotic errors in RPE-1 cells

To clarify the cause of chromosome segregation errors and mitotic delay in CENP-C$^{\Delta M12BD}$ RPE-1 cells, we first examined the K-fiber, as the levels of Ndc80C, which is a critical microtubule-binding complex, were significantly reduced in CENP-C$^{\Delta M12BD}$ RPE-1 cells. After calcium treatment to depolymerize the highly dynamic microtubules, we stained the remaining stable microtubules, which corresponded to K-fiber, and found that the K-fiber signal intensities in CENP-C$^{\Delta M12BD}$ RPE-1 cells were significantly lower than those in CENP-C$^{WT}$ RPE-1 cells, suggesting a reduction in K-fiber in CENP-C$^{\Delta M12BD}$ RPE-1 cells (Fig 3I). The CENP-C$^{\Delta M12BD}$ cells were more sensitive to low-dose nocodazole treatment than the CENP-C$^{WT}$ RPE-1 cells (Fig 3J). This result further supported the reduction of K-fiber in CENP-C$^{\Delta M12BD}$ RPE-1 cells.

Next, to examine whether the K-fiber reduction was the cause of mitotic defects in CENP-C$^{\Delta M12BD}$ RPE-1 cells, we reduced Ndc80C levels and consequently reduced K-fiber using CENP-T mutants lacking one Ndc80C-binding site. We generated CENP-T mutant RPE-1 cell lines, in which full-length human CENP-T fused with auxin-inducible degron (AID)-tag was expressed from the *AAVS1* locus, and *mScarlet-fused* mutant *human CENP-T* cDNAs lacking either one of two Ndc80C-binding sites (NBD-1: aa 6–31; NBD-2: aa 76–105) were introduced into the endogenous *CENP-T* locus (Figs 4A and S3A–E, *CENP-T$^{\Delta NBD-1}$* or *CENP-T$^{\Delta NBD-2}$*). Upon addition of auxin (IAA), AID-tagged CENP-T was degraded, and these cells expressed only the mScarlet-fused CENP-T mutant protein (Fig S3F).

In RPE-1 cells expressing CENP-T$^{\Delta NBD-1}$ (CENP-T$^{\Delta NBD-1}$ cells) or CENP-T$^{\Delta NBD-2}$ (CENP-T$^{\Delta NBD-2}$ RPE-1 cells), Ndc80C levels (Hec1) at the kinetochores were significantly lower than those in RPE-1 cells expressing WT CENP-T (CENP-T$^{WT}$ RPE-1 cells) (Fig 4B). Importantly, the Mis12C (DSN1) and KNL1C (KNL1) levels at the kinetochores in CENP-T$^{\Delta NBD-1}$ and CENP-T$^{\Delta NBD-2}$ RPE-1 cells were comparable to those in CENP-T$^{WT}$ RPE-1

**Figure 1. Mis12C-binding domain of CENP-C is dispensable for mouse development but is required for proper mitotic progression in MEFs.**
**(A)** Schematic representation of human and mouse CENP-C proteins. The Mis12-binding domain (M12BD) of human CENP-C and its homologous region in mouse CENP-C are highlighted in each CENP-C WT. To establish *Cenpc1* (*Cenpc*) mutant mice lacking the M12BD (*Cenpc$^{\Delta M12BD}$*), using CRISPR/Cas9 genome editing, exons 2–4 encoding amino acids 7–75 were deleted from the *Cenpc* gene locus (CENP-C ΔM12BD; see also Fig S1). **(B)** Genotype of offspring from *Cenpc$^{+/\Delta M12BD}$* intercross (chi-squared test, ***$P < 0.001$). **(C)** Growth curve of the MEFs isolated from *Cenpc$^{+/+}$*, *Cenpc$^{+/\Delta M12BD}$*, or *Cenpc$^{\Delta M12BD/\Delta M12BD}$* embryos. The cell numbers were normalized to those at day 0 of each line. Error bars indicate the mean and SD. Representative cell images at day 2 are also shown. Scale bar, 200 $\mu$m. **(D)** Representative time-lapse images of mitotic progression in *Cenpc$^{+/+}$*, *Cenpc$^{+/\Delta M12BD}$*, or *Cenpc$^{\Delta M12BD/\Delta M12BD}$* MEFs. DNA was visualized by staining with SPY650-DNA. Images were projected using maximum intensity projection and deconvolved. Time is relative to the nuclear envelope breakdown (NEBD). Scale bar, 10 $\mu$m. **(E)** Mitotic duration from the NEBD to the anaphase onset in *Cenpc$^{+/+}$*, *Cenpc$^{+/\Delta M12BD}$*, or *Cenpc$^{\Delta M12BD/\Delta M12BD}$* MEFs. The time-lapse images were analyzed to measure the time from the NEBD to the anaphase onset. Two independent experiments were performed (mean and SD, one-way ANOVA with Dunnett's multiple comparison test, exp1: n = 121 (*Cenpc$^{+/+}$*), 107 (*Cenpc$^{+/\Delta M12BD}$*), 117 (*Cenpc$^{\Delta M12BD/\Delta M12BD}$*); exp2: n = 121 (*Cenpc$^{+/+}$*), 124 (*Cenpc$^{+/\Delta M12BD}$*), 121 (*Cenpc$^{\Delta M12BD/\Delta M12BD}$*)). **(F)** Chromosome segregation errors in *Cenpc$^{+/+}$*, *Cenpc$^{+/\Delta M12BD}$*, or *Cenpc$^{\Delta M12BD/\Delta M12BD}$* MEFs. Lagging chromosomes and chromosome bridges during anaphase in the cells analyzed in (E) were scored. Representative images are shown. Scale bar, 10 $\mu$m. **(G)** Micronucleus formation in *Cenpc$^{+/+}$*, *Cenpc$^{+/\Delta M12BD}$*, or *Cenpc$^{\Delta M12BD/\Delta M12BD}$* MEFs. MEFs were fixed, and the interphase cells with micronuclei were scored (exp1: n = 1,570 [*Cenpc$^{+/+}$*], 1,753 [*Cenpc$^{+/\Delta M12BD}$*], 1,679 [*Cenpc$^{\Delta M12BD/\Delta M12BD}$*]; exp2: n = 818 [*Cenpc$^{+/+}$*], 754 [*Cenpc$^{+/\Delta M12BD}$*], 765 [*Cenpc$^{\Delta M12BD/\Delta M12BD}$*]). Representative images are shown. Scale bar, 10 $\mu$m. **(H)** Flow cytometry analysis of DNA contents in *Cenpc$^{+/+}$*, *Cenpc$^{+/\Delta M12BD}$*, or *Cenpc$^{\Delta M12BD/\Delta M12BD}$* MEFs. MEFs from exp1 were fixed and stained with propidium iodide and analyzed by flow cytometry. Representative results are shown on the left. Each cell line was tested in triplicate and analyzed (mean and SD, one-way ANOVA with Dunnett's multiple comparison test).

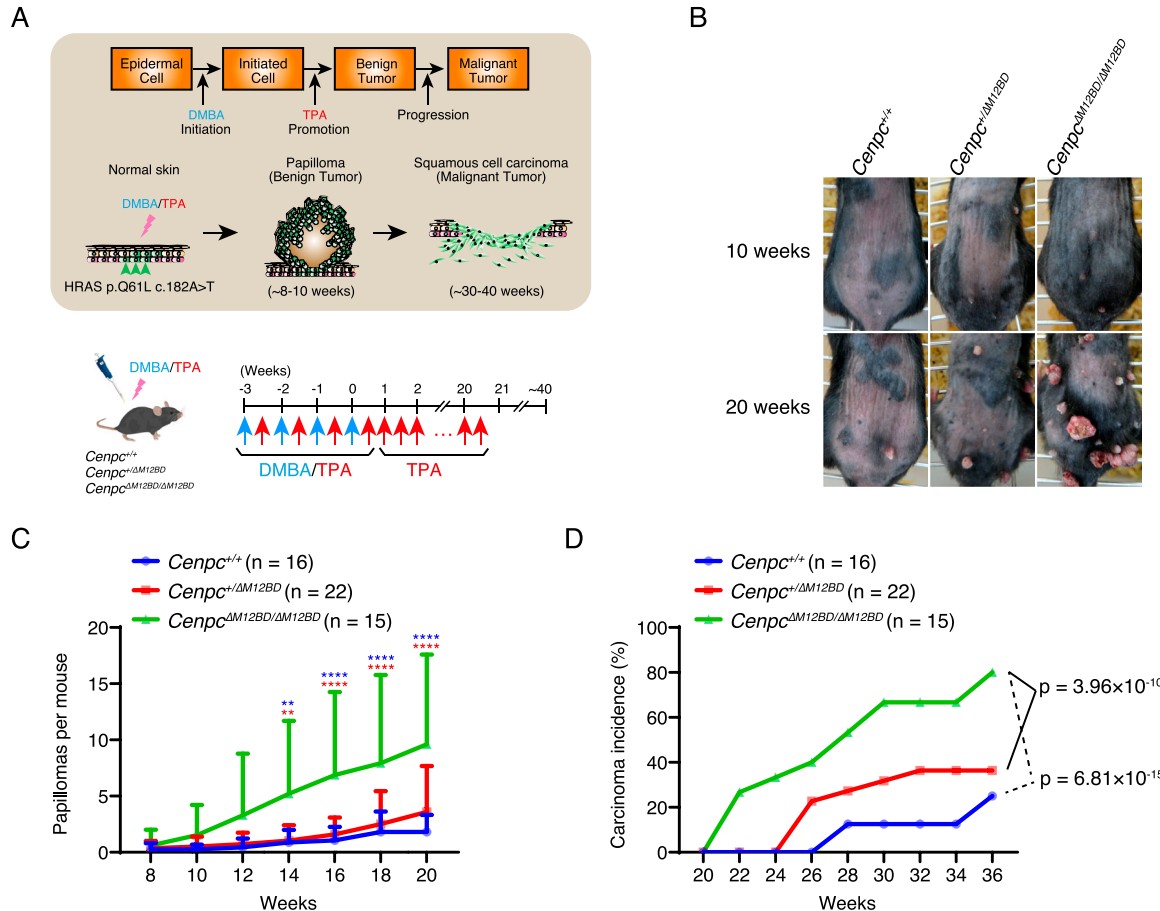

**Figure 2. Deletion of the Mis12C-binding domain of CENP-C exacerbates tumor formation and malignant conversion in a two-stage skin carcinogenesis model.**
**(A)** Schematic representation of the two-stage skin carcinogenesis model. 7,12-Dimethylbenz(a)anthracene (DMBA) and 12-O-tetradecanoylphorbol-13-acetate (TPA) were applied to the shaved dorsal back skin. After four rounds of DMBA/TPA treatment, the mice were further treated with TPA to promote papilloma formation and malignant conversion. The pipette and mouse images are adapted from Togo TV (https://togotv.dbcls.jp/en). **(B)** Representative images of papillomas on the back skin of *Cenpc*[+/+], *Cenpc*[+/ΔM12BD], or *Cenpc*[ΔM12BD/ΔM12BD] mice at 10 and 20 wk after the DMBA/TPA cycles. **(C)** Papilloma development in *Cenpc*[+/+], *Cenpc*[+/ΔM12BD], or *Cenpc*[ΔM12BD/ΔM12BD] mice. The numbers of papillomas on the back skin were counted for each mouse (mean and SD, two-way ANOVA with Tukey's test, **$P < 0.01$, ****$P < 0.0001$, blue: *Cenpc*[+/+] versus *Cenpc*[ΔM12BD/ΔM12BD]; red: *Cenpc*[+/ΔM12BD] versus *Cenpc*[ΔM12BD/ΔM12BD]). **(D)** Malignant conversion in *Cenpc*[+/+], *Cenpc*[+/ΔM12BD], or *Cenpc*[ΔM12BD/ΔM12BD] mice. Mice with squamous cell carcinoma were scored from 20 to 36 wk after the DMBA/TPA cycles (Fisher's exact test, *Cenpc*[+/+]: n = 16; *Cenpc*[+/ΔM12BD]: n = 22; *Cenpc*[ΔM12BD/ΔM12BD]: n = 15).

cells (Fig 4C and D). As Ndc80C was reduced at the kinetochores, the K-fiber signal intensities in CENP-T[ΔNBD−1] or CENP-T[ΔNBD−2] RPE-1 cells were significantly lower than those in CENP-T[WT] RPE-1 cells. Consistent with K-fiber reduction, CENP-T[ΔNBD−1] and CENP-T[ΔNBD−2] RPE-1 cells showed sensitivity to low-dose nocodazole (Fig 4F).

Despite the reduced Ndc80C and associated K-fiber reduction, CENP-T[ΔNBD−1] and CENP-T[ΔNBD−2] RPE-1 cells showed neither significant mitotic delay nor increased chromosome segregation errors (Fig 4G–I). These results suggest that the K-fiber reduction because of the decrease in Ndc80C is not the cause of mitotic defects in CENP-C[ΔM12BD] RPE-1 cells.

### Aurora B localization to mitotic centromeres is diminished in CENP-C[ΔM12BD] RPE-1 cells

To determine the cause of chromosome segregation errors and mitotic delay in CENP-C[ΔM12BD] RPE-1 cells, we examined the regulatory mechanisms of kinetochore–microtubule attachment. Aurora

B is a conserved mitotic kinase that phosphorylates kinetochore substrates, such as Hec1, facilitating the correction of erroneous kinetochore–microtubule attachment (Cheeseman et al, 2006; DeLuca et al, 2006; Liu et al, 2009; DeLuca et al, 2011; Zaytsev et al, 2014; Long et al, 2017). As shown in Fig 5A, Aurora B was localized to the inner centromeric region between sister kinetochores in mitotic cells. We found that Aurora B levels were significantly reduced in CENP-C[ΔM12BD] RPE-1 cells compared with those in CENP-C[WT] RPE-1 cells (Fig 5A). In contrast, the reduction in Aurora B at the centromeres was not observed in CENP-T[ΔNBD−1] or CENP-T[ΔNBD−2] RPE-1 cells (Fig 5B), which underwent proper mitotic progression despite the reduction of Ndc80C levels at kinetochores (Fig 4).

To further evaluate the reduction of Aurora B activity at the centromeres in CENP-C[ΔM12BD] RPE-1 cells, we examined metaphase chromosome oscillation, which is regulated by Aurora B through the phosphorylation of Hec1 (DeLuca et al, 2011; Zaytsev et al, 2014; Long et al, 2017). CENP-C[WT] or CENP-C[ΔM12BD] RPE-1 cells were treated with MG132, a proteasome inhibitor, to arrest the cells at metaphase, and

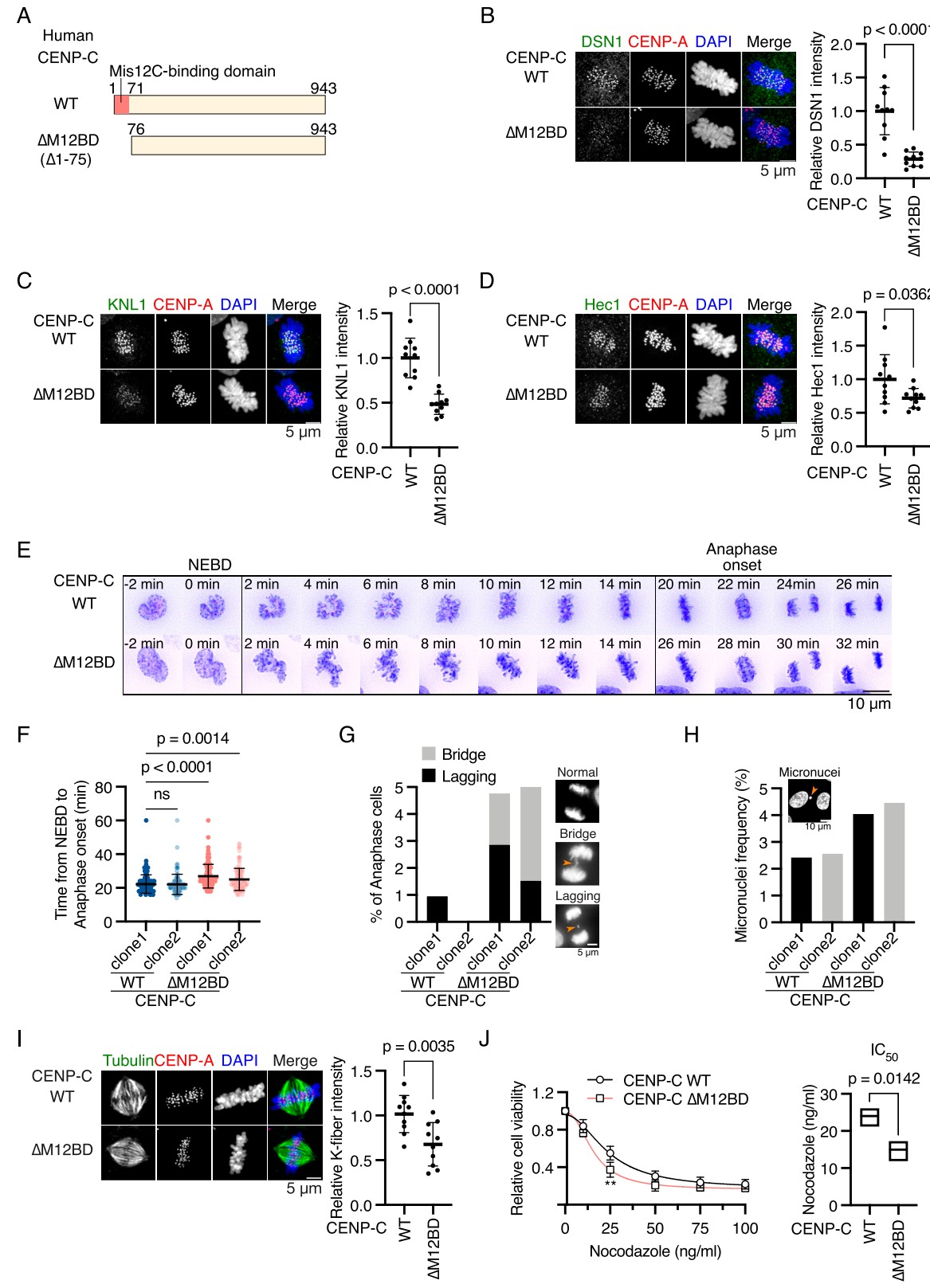

**Figure 3. Deletion of the Mis12C-binding domain of human CENP-C delays mitotic progression and increases chromosome segregation errors in RPE-1 cells.**
**(A)** Schematic representation of human CENP-C. The Mis12C-binding domain (M12BD, amino acids 1–71) is highlighted in CENP-C WT. The N-terminal region (amino acids 1–75) was deleted in CENP-C$^{\Delta M12BD}$. *FLAG-tagged CENP-C WT or ΔM12BD* was introduced into the *CENP-C* locus in RPE-1 cells expressing mScarlet-CENP-A and GFP-H2A (CENP-C$^{WT}$ or CENP-C$^{\Delta M12BD}$ RPE-1 cells, respectively; see Fig S2). **(B)** DSN1 localization in CENP-C$^{WT}$ or CENP-C$^{\Delta M12BD}$ RPE-1 cells. DSN1 was stained with an antibody against

chromosome oscillations were observed by time-lapse imaging (Fig S4A). The oscillation amplitude was assessed by quantifying the deviation from the average position (DAP) (Stumpff et al, 2008) for tracked kinetochores labeled with mScarlet-CENP-A (Fig S4B and C). The amplitude in CENP-C$^{\Delta M12BD}$ RPE-1 cells was significantly smaller than that in CENP-C$^{WT}$ RPE-1 cells (Fig 5C). In contrast, CENP-T$^{NBD-1}$ and CENP-T$^{NBD-2}$ RPE-1 cells did not show changes in the amplitude of oscillations compared with CENP-T$^{WT}$ RPE-1 cells (Figs 5C and S4D). These results suggest a reduction in Aurora B activity at the centromeres in CENP-C$^{\Delta M12BD}$ RPE-1 cells.

Centromeric localization of Aurora B is promoted by the phosphorylation of histone H3 at threonine 3 (H3T3ph) by Haspin and histone H2A at threonine 120 (H2AT120ph) by Bub1 (Watanabe, 2010; Wang et al, 2011). To investigate how the CENP-C-Mis12C interaction is related to Aurora B localization, we examined the phosphorylation of these histones and found that H2AT120ph levels were significantly reduced in CENP-C$^{\Delta M12BD}$ RPE-1 cells; however, H3T3ph levels in CENP-C$^{\Delta M12BD}$ RPE-1 cells were comparable to those in CENP-C$^{WT}$ RPE-1 cells (Fig 5D and E). Consistent with the changes in H2AT120ph, CENP-C$^{\Delta M12BD}$ RPE-1 cells showed less Bub1 localization at the kinetochores during mitotic progression than did CENP-C$^{WT}$ cells (Fig 5F). Because Bub1 localizes to kinetochores through KNL1 (Kiyomitsu et al, 2007), the results aligned with the reduction of KNL1C, which binds to Mis12C (Petrovic et al, 2010, 2014, 2016) at kinetochores in CENP-C$^{\Delta M12BD}$ RPE-1 cells (Fig 3C).

As KMN recruits many other proteins, we also examined the localization of known kinetochore-localizing proteins, which contribute to kinetochore–microtubule attachment and chromosome segregation: Ska complex (Ska3), BubR1, and PLK1 (Hanisch et al, 2006; Welburn et al, 2009; Chan et al, 2012; Overlack et al, 2015; Ikeda & Tanaka, 2017; Chen et al, 2021; Singh et al, 2021). The kinetochore localization of these proteins was reduced in CENP-C$^{\Delta M12BD}$ RPE-1 cells (Fig S5A–C). The Ska complex interacts with Ndc80C to bind microtubules (Zhang et al, 2017). As Ndc80C reduction did not cause mitotic defects (Fig 4), the Ska complex reduction would not be a major reason for mitotic defects in CENP-C$^{\Delta M12BD}$ RPE-1 cells. BubR1 and PLK1, as well as Bub1, are key regulators of the spindle assembly checkpoint (SAC). Although we cannot rule out the possibility that SAC reduction is a cause of mitotic defects, because CENP-C$^{\Delta M12BD}$ RPE-1 cells showed unaligned chromosomes during

prometaphase (Fig 3E), we further investigated Aurora B, which regulates kinetochore–microtubule attachment during prometaphase, and its function.

In contrast to reduction of Bub1, H2AT120ph, and Aurora B levels in CENP-C$^{\Delta M12BD}$ RPE-1 cells, those levels in CENP-T$^{\Delta NBD-1}$ and CENP-T$^{\Delta NBD-2}$ RPE-1 cells, which showed neither mitotic defects nor Aurora B reduction, were comparable to those in CENP-T$^{WT}$ RPE-1 cells (Figs 5B and S6). The unaltered Bub1 and H2AT120ph levels were consistent with the finding that Mis12C and KNL1C levels at the centromeres in CENP-T$^{\Delta NBD-1}$ and CENP-T$^{\Delta NBD-2}$ RPE-1 cells were comparable to those in CENP-T$^{WT}$ RPE-1 cells (Fig 4C and D).

These data suggest that the CENP-C-Mis12C interaction positively regulates centromeric Aurora B localization through the Bub1/H2AT120ph axis.

### Centromeric Aurora B localization is reduced in CENP-T$^{\Delta M12BD}$ RPE-1 cells

Although we focused on Mis12C binding to CENP-C, Mis12C also binds to another CCAN protein CENP-T. To examine levels of Mis12C on CENP-T, we generated RPE-1 cells in which endogenous CENP-T was replaced with mScarlet-human CENP-T lacking the Mis12C-binding domain (aa 107–230 region; M12BD) and *AID-fused CENP-T* was introduced into the *AAVS* locus (Fig S7A–E). Once IAA was added, only mutant CENP-T was expressed in these cells, which were referred to as CENP-T$^{\Delta M12BD}$ RPE-1 cells (Fig S7A–E). First, we tested DSN1 localization in CENP-T$^{\Delta M12BD}$ RPE-1 cells. As observed in CENP-C$^{\Delta M12BD}$ RPE-1 cells, DSN1 levels were half in CENP-T$^{\Delta M12BD}$ RPE-1 cells, compared with those in CENP-T$^{WT}$ RPE-1 cells (Fig S7F). We also observed a reduction of KNL1, Hec1, Bub1, and H2AT120ph levels in CENP-T$^{\Delta M12BD}$ RPE-1 cells (Fig S7G–J). Consistent with these observations, Aurora B levels were also reduced in CENP-T$^{\Delta M12BD}$ RPE-1 cells (Fig S7K), suggesting that Mis12C on CENP-T also contributes to proper mitotic progression.

### Deletion of the DSN1 basic motif enhanced Mis12C levels on CENP-C but not on CENP-T in RPE-1 cells

The CENP-C-Mis12C interaction is regulated by the DSN1 basic motif, which masks the Mis12C surface for CENP-C binding in the

---

DSN1 (green). mScarlet-CENP-A was used as a kinetochore marker (CENP-A, red). DNA was stained with DAPI (blue). Scale bar, 5 $\mu$m. DSN1 signal intensities at mitotic kinetochores were quantified (mean and SD, two-tailed *t* test, CENP-C$^{WT}$ cells: n = 10; CENP-C$^{\Delta M12BD}$ RPE-1 cells: n = 10). **(C)** KNL1 localization in CENP-C$^{WT}$ or CENP-C$^{\Delta M12BD}$ RPE-1 cells. KNL1 was stained with an antibody against KNL1. KNL1 localization at mitotic kinetochores was examined and quantified as in (B). Scale bar, 5 $\mu$m. Mean and SD, two-tailed *t* test, CENP-C$^{WT}$ cells: n = 10; CENP-C$^{\Delta M12BD}$ cells: n = 10. **(D)** Hec1 localization in CENP-C$^{WT}$ or CENP-C$^{\Delta M12BD}$ RPE-1 cells. Hec1 was stained with an antibody against Hec1. Hec1 localization at mitotic kinetochores was examined and quantified as in (B). Scale bar, 5 $\mu$m. Mean and SD, two-tailed *t* test, CENP-C$^{WT}$ cells: n = 10; CENP-C$^{\Delta M12BD}$ RPE-1 cells: n = 10. **(E)** Representative time-lapse images of mitotic progression in CENP-C$^{WT}$ or CENP-C$^{\Delta M12BD}$ cells. DNA was visualized with GFP-H2A. Images were projected using maximum intensity projection and deconvolved. Time is relative to the nuclear envelope breakdown (NEBD). Scale bar, 10 $\mu$m. **(F)** Mitotic duration from the NEBD to the anaphase onset in CENP-C$^{WT}$ or CENP-C$^{\Delta M12BD}$ cells. The time-lapse images were analyzed to measure the time from the NEBD to the anaphase onset. Two independent clones of CENP-C$^{WT}$ or CENP-C$^{\Delta M12BD}$ RPE-1 cells were tested (mean and SD, two-tailed *t* test, CENP-C$^{WT}$ RPE-1 cell clone1: n = 120; CENP-C$^{WT}$ RPE-1 cell clone2: n = 105; CENP-C$^{\Delta M12BD}$ RPE-1 cell clone1: n = 105; CENP-C$^{\Delta M12BD}$ RPE-1 cell clone2: n = 132). **(G)** Chromosome segregation errors in CENP-C$^{WT}$ or CENP-C$^{\Delta M12BD}$ RPE-1 cells. Lagging chromosomes and chromosome bridges during anaphase in the cells analyzed in (F) were scored. Representative images are shown. Scale bar, 5 $\mu$m. **(H)** Micronucleus formation in CENP-C$^{WT}$ or CENP-C$^{\Delta M12BD}$ RPE-1 cells. The cells were fixed, and the interphase cells with micronuclei were scored (CENP-C$^{WT}$ RPE-1 cell clone1: n = 1,527; CENP-C$^{WT}$ RPE-1 cell clone2: n = 976; CENP-C$^{\Delta M12BD}$ RPE-1 cell clone1: n = 1,633; CENP-C$^{\Delta M12BD}$ RPE-1 cell clone2: n = 1,076). Scale bar, 10 $\mu$m. **(I)** K-fiber in CENP-C$^{WT}$ or CENP-C$^{\Delta M12BD}$ RPE-1 cells. CENP-C$^{WT}$ or CENP-C$^{\Delta M12BD}$ RPE-1 cells expressing mScarlet CENP-A were fixed after CaCl$_2$ treatment and stained with an anti-alpha-tubulin antibody. Scale bar, 5 $\mu$m. The means of tubulin signal intensities of the spindle in a cell were quantified as K-fiber signals (mean and SD, two-tailed *t* test, CENP-C$^{WT}$ RPE-1 cells: n = 10; CENP-C$^{\Delta M12BD}$ RPE-1 cells: n = 10). **(J)** Cell viability of CENP-C$^{WT}$ or CENP-C$^{\Delta M12BD}$ RPE-1 cells treated with various concentrations of nocodazole. Viable cells were measured 3 d after nocodazole addition. Three independent experiments were performed (mean and SD, two-way ANOVA with Šídák's multiple comparison test, **$P$ = 0.0022). IC$_{50}$ indicates the average of concentration to reduce cell viability to 50% from three independent experiments (mean and SD, two-tailed *t* test).

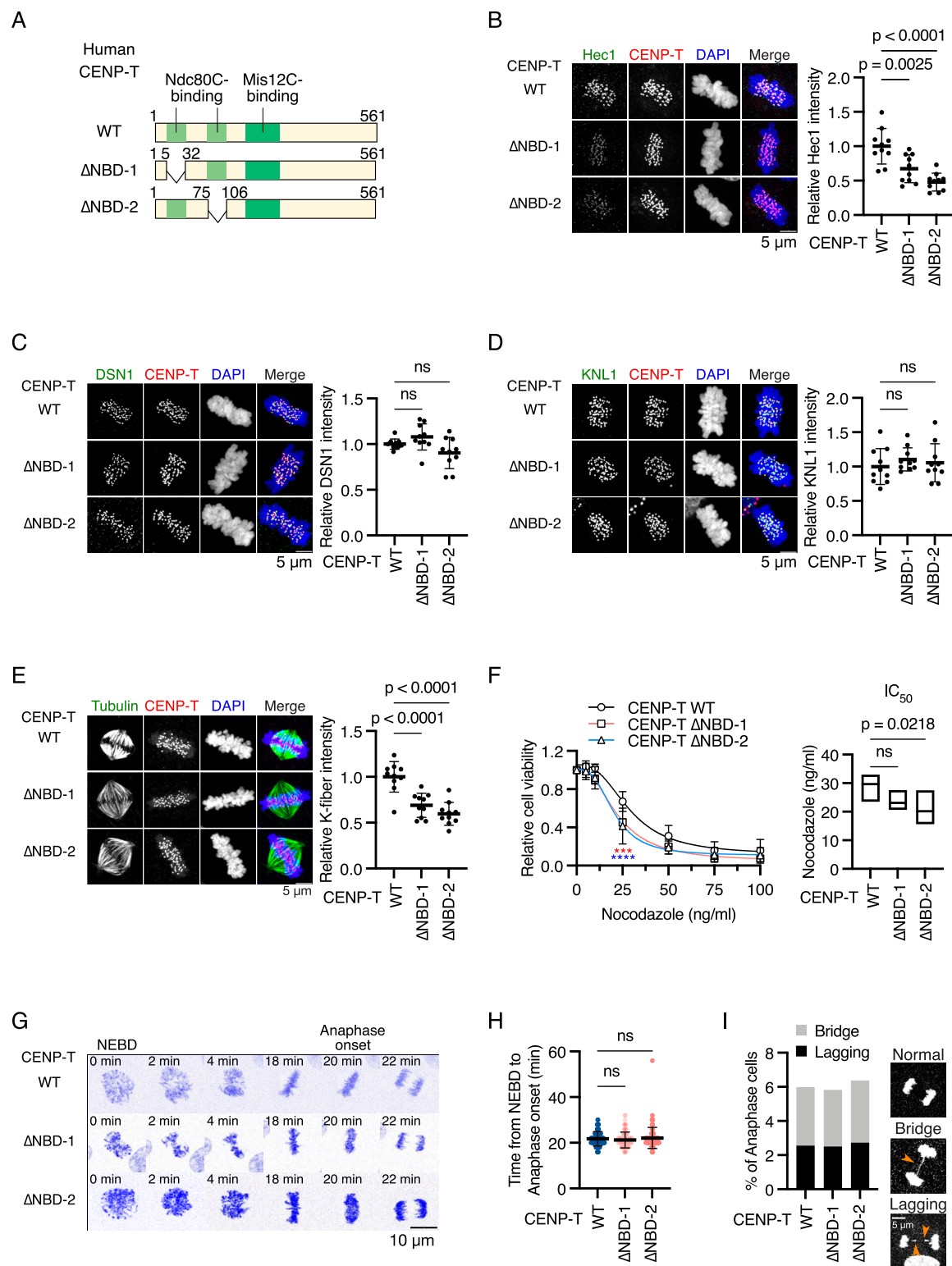

**Figure 4. Deletion of one Ndc80C-binding domain of human CENP-T reduces Ndc80C localization and K-fiber but does not cause mitotic defects in RPE-1 cells.**
**(A)** Schematic representation of human CENP-T. Human CENP-T WT has two Ndc80C-binding regions (NBD-1 or NBD-2: amino acids 6–31 or 76–105) and a Mis12-binding domain. Each NBD was deleted in CENP-T$^{ΔNBD-1}$ or CENP-T$^{ΔNBD-2}$ RPE-1 cells, respectively. *mScarlet-fused CENP-T WT* or each mutant was introduced into the *CENP-T* locus in RPE-1 cells expressing OsTIR1 and GFP-mAID-CENP-T (CENP-T$^{WT}$, CENP-T$^{ΔNBD-1}$, or CENP-T$^{ΔNBD-2}$ RPE-1 cells, respectively; see Fig S3). **(B)** Hec1 localization in CENP-T$^{WT}$, CENP-T$^{ΔNBD-1}$, or CENP-T$^{ΔNBD-2}$ RPE-1 cells. Hec1 was stained with an antibody against Hec1 (green). mScarlet-CENP-T was used as a kinetochore marker (CENP-T, red). DNA was stained with DAPI (blue). Scale bar, 5 *μ*m. Hec1 signal intensities at mitotic kinetochores were quantified (mean and SD, one-way ANOVA with Dunnett's multiple comparison test, CENP-T$^{WT}$ RPE-1 cells: n = 10; CENP-T$^{ΔNBD-1}$ RPE-1 cells: n = 10; CENP-T$^{ΔNBD-2}$ RPE-1 cells: n = 10). **(C)** DSN1 localization in CENP-T$^{WT}$, CENP-T$^{ΔNBD-1}$, or CENP-

nonphosphorylation state of DSN1, and Aurora B phosphorylation of the basic motif releases it from the Mis12C surface to facilitate the CENP-C-Mis12C interaction (Kim & Yu, 2015; Rago et al, 2015; Dimitrova et al, 2016; Petrovic et al, 2016; Hara et al, 2018). The enhanced binding of CENP-C to Mis12C was also observed by DSN1 basic motif deletion. Recent studies also suggest that the DSN1 basic motif might also regulate the CENP-T-Mis12C interaction (Walstein et al, 2021; Polley et al, 2024; Yatskevich et al, 2024). However, these experiments were mainly performed in vitro, and it was unknown how mitotic kinetochore localization of Mis12C is regulated by the DSN1 basic motif in cells. Therefore, we tested how Mis12C levels were changed after DSN1 basic motif deletion in CENP-C$^{WT}$ or CENP-C$^{ΔM12BD}$ RPE-1 cells. First, we generated a CENP-C$^{WT}$ or CENP-C$^{ΔM12BD}$ RPE-1 cell expressing a mScarlet-fused DSN1 mutant lacking the basic motif (Δ91–113 aa) from the endogenous *DSN1* locus (CC$^{WT}$/DSN1$^{ΔBM}$ or CC$^{ΔM12BD}$/DSN1$^{ΔBM}$ RPE-1 cells; Fig S8A–C). We also introduced mScarlet-fused WT DSN1 into CENP-C$^{WT}$ or CENP-C$^{ΔM12BD}$ RPE-1 cells (CC$^{WT}$/DSN1$^{WT}$ or CC$^{ΔM12BD}$/DSN1$^{WT}$ RPE-1 cells). When we compared Mis12C (DSN1) levels of CC$^{WT}$/DSN1$^{WT}$ with those of CC$^{WT}$/DSN1$^{ΔBM}$ RPE-1 cells, those at kinetochores, as well as KNL1C (KNL1) and Bub1 levels, were increased in CC$^{WT}$/DSN1$^{ΔBM}$ RPE-1 cells compared with those in CC$^{WT}$/DSN1$^{WT}$ cells (Fig S8D–F). However, when we compared those of CC$^{ΔM12BD}$/DSN1$^{WT}$ to CC$^{ΔM12BD}$/DSN1$^{ΔBM}$ RPE-1 cells, DSN1, KNL1, and Bub1 levels were comparable (Fig S8D–F). These results indicate Mis12C on CENP-C was increased, but not on CENP-T, after Dsn1 basic motif deletion. This might be explained by a previous observation, in which almost all CENP-T molecules at kinetochores associate with Mis12C, but a large pool of CENP-C is Mis12C-free in the WT cells (Suzuki et al, 2015). Consistent with increased Mis12C, Aurora B levels at centromeres were increased in CC$^{WT}$/DSN1$^{ΔBM}$ RPE-1 cells compared with those in CC$^{WT}$/DSN1$^{WT}$ RPE-1 cells (Fig S8G). However, those are not changed between CC$^{ΔM12BD}$/DSN1$^{WT}$ and CC$^{ΔM12BD}$/DSN1$^{ΔBM}$ RPE-1 cells (Fig S8G). As Bub1 regulates Aurora B localization at the kinetochore-proximal region through H2AT120ph, we examined levels of the kinetochore-proximal pool of Aurora B using a Haspin inhibitor (Broad et al, 2020). The Aurora B levels at the kinetochore-proximal region have similar profiles to those at centromeres (Fig S8H).

## Error correction efficiency was reduced in CENP-C$^{ΔM12BD}$ RPE-1 cells

As one of the key functions of Aurora B is mitotic error correction, which resolves erroneous kinetochore–microtubule attachments to facilitate the correct bipolar spindle microtubule attachment to the sister kinetochores (Dewar et al, 2004; Lampson et al, 2004; Cimini et al, 2006), we evaluated the error correction efficiency in CENP-C$^{ΔM12BD}$ RPE-1 cells with low Aurora B levels at centromeres (Fig 6A). RPE-1 cells were treated with monastrol, a reversible Eg5 inhibitor (Kapoor et al, 2000), to induce monopolar spindles with erroneous kinetochore–microtubule attachment. After release from monastrol, the cells were arrested at metaphase with MG132, and cells with unaligned chromosomes were scored (Lampson et al, 2004) (Fig 6A). As a control for this error correction assay, we treated CENP-C$^{WT}$ RPE-1 cells with AZD1152, an Aurora B inhibitor, and confirmed an increase in cells with unaligned chromosomes 30 min after release (over 10% and 20% with 40 and 200 nM AZD1152, respectively), compared to the cells treated with DMSO (less than 10%) (Fig 6A). This observation is consistent with previous reports, demonstrating that efficient error correction requires Aurora B (Lampson et al, 2004; Cimini et al, 2006).

Next, we examined CENP-C$^{ΔM12BD}$ RPE-1 cells and found that the cells with unaligned chromosomes were significantly increased to ~20% 30 min after release from monastrol (Fig 6A), suggesting that error correction was less efficient in CENP-C$^{ΔM12BD}$ RPE-1 cells and that the M12BD plays a part in mitotic error correction to facilitate bipolar attachment. In contrast, CENP-T$^{ΔNBD-1}$ and CENP-T$^{ΔNBD-2}$ RPE-1 cells exhibited error correction efficiencies equivalent to those in CENP-T$^{WT}$ RPE-1 cells (Fig 6B). This is consistent with the fact that the deletion of either NBD-1 or NBD-2 did not alter the localization of centromeric Aurora B (Fig 5B). In addition, the results implied that reduced K-fiber and Ndc80C levels did not affect error correction efficiency in this assay.

These results suggest that deletion of the M12BD of CENP-C diminishes error correction, possibly because of the reduction of Aurora B levels at centromeres, which is the primary cause of mitotic delay and chromosome segregation errors in CENP-C$^{ΔM12BD}$ RPE-1 cells.

---

T$^{ΔNBD-2}$ RPE-1 cells. DSN1 was stained with an antibody against DSN1 (green). DSN1 localization at mitotic kinetochores was examined and quantified as in (B). Scale bar, 5 $\mu$m. Mean and SD, one-way ANOVA with Dunnett's multiple comparison test, CENP-T$^{WT}$ cells: n = 10; CENP-T$^{ΔNBD-1}$ RPE-1 cells: n = 10; CENP-T$^{ΔNBD-2}$ cells: n = 10. **(D)** KNL1 localization in CENP-T$^{WT}$, CENP-T$^{ΔNBD-1}$, or CENP-T$^{ΔNBD-2}$ RPE-1 cells. KNL1 was stained with an antibody against KNL1 (green). KNL1 localization at mitotic kinetochores was examined and quantified as in (B). Scale bar, 5 $\mu$m. Mean and SD, one-way ANOVA with Dunnett's multiple comparison test, CENP-T$^{WT}$ RPE-1 cells: n = 10; CENP-T$^{ΔNBD-1}$ RPE-1 cells: n = 10; CENP-T$^{ΔNBD-2}$ RPE-1 cells: n = 10. **(E)** K-fiber in CENP-T$^{WT}$, CENP-T$^{ΔNBD-1}$, or CENP-T$^{ΔNBD-2}$ RPE-1 cells. CENP-T$^{WT}$, CENP-T$^{ΔNBD-1}$, or CENP-T$^{ΔNBD-2}$ RPE-1 cells were fixed after CaCl$_2$ treatment and stained with an anti-alpha-tubulin antibody (green). CENP-T fused with mScarlet was used as a kinetochore marker (CENP-T, red). Scale bar, 5 $\mu$m. The means of tubulin signal intensities of the spindle in a cell were quantified as K-fiber signals (mean and SD, one-way ANOVA with Dunnett's multiple comparison test, CENP-T$^{WT}$ RPE-1 cells: n = 10; CENP-T$^{ΔNBD-1}$ RPE-1 cells: n = 10; CENP-T$^{ΔNBD-2}$ RPE-1 cells: n = 10). **(F)** Cell viability of CENP-T$^{WT}$, CENP-T$^{ΔNBD-1}$, or CENP-T$^{ΔNBD-2}$ RPE-1 cells treated with various concentrations of nocodazole. Viable cells were measured 3 d after nocodazole addition. Four independent experiments were performed (mean and SD, two-way ANOVA with Dunnett's multiple comparison test, ***$P$ = 0.0007, ****$P$ < 0.0001, red: WT versus ΔNBD-1; blue: WT versus ΔNBD-2). IC$_{50}$ indicates the average of nocodazole concentration to reduce cell viability to 50% from four independent experiments (mean and SD, one-way ANOVA with Dunnett's multiple comparison test). **(G)** Representative time-lapse images of mitotic progression in CENP-T$^{WT}$, CENP-T$^{ΔNBD-1}$, or CENP-T$^{ΔNBD-2}$ RPE-1 cells. DNA was visualized with SPY505-DNA. Images were projected using maximum intensity projection and deconvolved. Time is relative to the nuclear envelope breakdown (NEBD). Scale bar, 10 $\mu$m. **(H)** Mitotic duration from the NEBD to the anaphase onset in CENP-T$^{WT}$, CENP-T$^{ΔNBD-1}$, or CENP-T$^{ΔNBD-2}$ RPE-1 cells. The time-lapse images were analyzed to measure the time from the NEBD to the anaphase onset (mean and SD, one-way ANOVA with Dunnett's multiple comparison test, CENP-T$^{WT}$ RPE-1 cells: n = 117; CENP-T$^{ΔNBD-1}$ RPE-1 cells: n = 120; CENP-T$^{ΔNBD-2}$ RPE-1 cells: n = 110). **(I)** Chromosome segregation errors in CENP-T$^{WT}$, CENP-T$^{ΔNBD-1}$, or CENP-T$^{ΔNBD-2}$ RPE-1 cells. Lagging chromosomes and chromosome bridges during anaphase in the cells analyzed in (H) were scored. Representative images are shown. Scale bar, 5 $\mu$m.

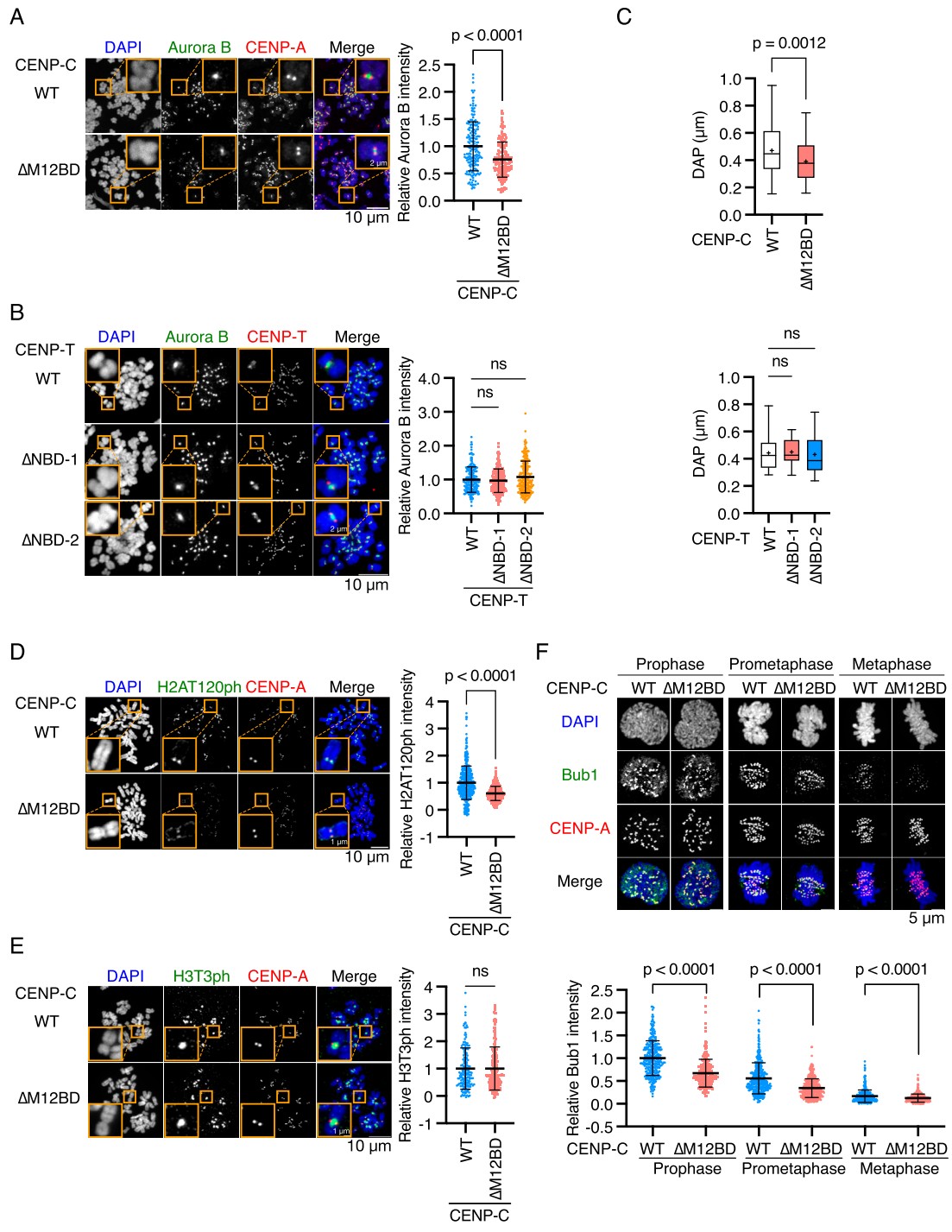

**Figure 5. Deletion of the Mis12C-binding domain of human CENP-C reduces Aurora B and Bub1 levels at centromeres in RPE-1 cells.**
**(A)** Aurora B localization in CENP-C^WT or CENP-C^ΔM12BD RPE-1 cells. Aurora B was stained with an antibody against Aurora B (green). mScarlet-CENP-A was used as a kinetochore marker (CENP-A, red). DNA was stained with DAPI (blue). Scale bar, 10 μm. The insets show an enlarged single chromosome (scale bar, 2 μm). Aurora B signal intensities at inner centromeres were quantified (mean and SD, two-tailed t test, CENP-C^WT RPE-1 cells: n = 188 centromeres from 5 cells; CENP-C^ΔM12BD RPE-1 cells: n = 166 centromeres from 5 cells). **(B)** Aurora B localization in CENP-T^WT, CENP-T^ΔNBD−1, or CENP-T^ΔNBD−2 RPE-1 cells. Aurora B and DNA were stained as in (A). mScarlet-CENP-T was used as a kinetochore marker (CENP-T, red). Scale bar, 5 μm. The Aurora B signal intensities were quantified (mean and SD, one-way ANOVA with Dunnett's multiple comparison test, CENP-T^WT RPE-1 cells: n = 197 centromeres from 5 cells; CENP-T^ΔNBD−1 RPE-1 cells: n = 198 centromeres from 5 cells; CENP-T^ΔNBD−2 RPE-1 cells: n = 225 centromeres from 5 cells). **(C)** Chromosome oscillation in CENP-C^WT, CENP-C^ΔM12BD, CENP-T^WT, CENP-T^ΔNBD−1, or CENP-T^ΔNBD−2 RPE-1 cells. The deviation from the average position (DAP) was calculated from the time-lapse images (Fig S4). The graphs display the median and quantile with max and min (+ indicates mean) (two-tailed t test, CENP-C^WT RPE-1 cells: n = 54 kinetochore pairs from 12 cells; CENP-C^ΔM12BD RPE-1 cells: n = 42 kinetochore pairs from 7 cells; one-way ANOVA with Dunnett's multiple comparison test, CENP-T^WT RPE-1 cells: n = 17 kinetochore pairs from 3 cells; CENP-T^ΔNBD−1 RPE-1 cells: n = 15 kinetochore pairs from 3 cells; CENP-T^ΔNBD−2 RPE-1 cells: n = 16 kinetochore pairs from 4 cells). **(D)** H2AT120ph localization in CENP-C^WT or CENP-C^ΔM12BD RPE-1 cells. H2AT120ph was stained with an antibody against H2AT120ph (green).

## Forced binding of Mis12C to CENP-C suppresses chromosomal instability in human HeLa cells

The aforementioned results led to a model in which the CENP-C-Mis12C interaction positively regulates Aurora B localization at centromeres, facilitating the biorientation of chromosomes. To prove this idea, we used HeLa cells. In contrast to the near-diploid RPE-1 cells, which have robust error correction and chromosome stability, HeLa cells, a cancer cell line with chromosomal instability, have low Aurora B activity at mitotic centromeres and inefficient error correction (Fig 7A) (Abe et al, 2016). We assumed that HeLa cells were an appropriate model to examine whether forcing Mis12C binding to CENP-C would increase Aurora B levels at the centromeres and ameliorate error correction efficiency (Fig 7A). To increase the Mis12C-CENP-C interaction, we expressed the DSN1 basic motif in HeLa cells (Fig 7A), as we have done in RPE-1 cells (Fig S8).

We generated a HeLa cell line expressing a DSN1 mutant lacking the basic motif (referred to as the DSN1$^{\Delta BM}$) from the endogenous *DSN1* locus (DSN1$^{\Delta BM}$ HeLa cells; Fig S9A–C). As observed in RPE-1 cells, Mis12C (DSN1) levels at kinetochores, as well as KNL1C (KNL1) and Ndc80C (Hec1) levels, were increased in DSN1$^{\Delta BM}$ HeLa cells compared with those in DSN1$^{WT}$ HeLa cells (Figs 7B and S9D), indicating that a stable CENP-C-Mis12C interaction occurred in Dsn1$^{\Delta BM}$ HeLa cells. This forced binding of Mis12C to CENP-C significantly increased the centromeric localization of Aurora B, Bub1, and H2AT120ph in DSN1$^{\Delta BM}$ HeLa cells (Fig 7C and D). Finally, we assessed the error correction efficiency in DSN1$^{\Delta BM}$ HeLa cells and found that it was improved in DSN1$^{\Delta BM}$ HeLa cells (Fig 7E). These results indicate that forced binding of Mis12C to CENP-C facilitates Aurora B localization at the centromeres and improves error correction efficiency in HeLa cells, supporting the hypothesis that the CENPC-Mis12C interaction positively regulates Aurora B localization at the centromeres.

# Discussion

In this study, we demonstrated that the CENP-C-Mis12C interaction positively regulates Aurora B localization and facilitates mitotic error correction to establish bioriented chromosomes. Given that Aurora B regulates CENP-C-Mis12C binding during mitosis through DSN1 phosphorylation, we propose a feedback mechanism to control Aurora B levels at the centromeres during mitosis (Fig 7F). Communication between the inner centromere and the outer kinetochore through CENP-C is crucial for maintaining sufficient levels of Aurora B for bioriented chromosome establishment and

accurate chromosome segregation to prevent chromosomal instability and cancer formation (Fig 7F).

The CENP-C-Mis12C interaction has been studied as a possible crucial platform for recruiting Ndc80C for microtubule binding, and its regulatory mechanisms have been revealed at the molecular level (Dimitrova et al, 2016; Petrovic et al, 2016). Contrary to its proposed importance, *Cenpc$^{\Delta M12BD/\Delta M12BD}$* mice are viable, suggesting that the CENP-C-Mis12C interaction is largely dispensable for mouse development. This is likely because CENP-T is the major scaffold for KMN recruitment to kinetochores in mammalian cells (MEFs and RPE-1 cells), as in chicken DT40 cells (Hara et al, 2018).

Nevertheless, *Cenpc$^{\Delta M12BD/\Delta M12BD}$* mice are cancer-prone in a two-stage skin carcinogenesis model. This observation suggests that the CENP-C-Mis12C interaction is of physiological importance for ensuring accurate chromosome segregation in cells exposed to stress. We demonstrated that MEFs established from *Cenpc$^{\Delta M12BD/\Delta M12BD}$* embryos showed significant chromosome segregation errors. Our culture conditions were not optimal for MEFs (Parrinello et al, 2003), resulting in high basal chromosome segregation errors even in control cells. We also found that the CENP-C-Mis12C interaction was unnecessary for cell proliferation in RPE-1 cells, a near-diploid human cell line; however, it was required for efficient bioriented chromosome establishment. Therefore, RPE-1 is a good model to examine functional roles of the CENP-C-Mis12C interaction.

We previously reported that the CENP-C-Mis12C interaction is dispensable for chicken DT40 cell proliferation (Hara et al, 2018). In contrast to mammalian cells (MEFs and RPE-1 cells), we did not observe any obvious mitotic defects in DT40 cells lacking the M12BD. This may be attributed to technical limitations in detecting subtle differences in chromosome segregation in DT40 cells, which have many tiny chromosomes in small cell volumes. An alternative explanation may be the difference in CENP-C characteristics between mammals and chickens. In human cells, CENP-C depletion causes CCAN disassembly (McKinley et al, 2015), whereas CCAN proteins, including CENP-T, remain on the centromeres in CENP-C-knockout chicken DT40 cells (Hori et al, 2008), implying that other kinetochore proteins compensate CENP-C functions for kinetochore assembly in chicken DT40 cells.

Error correction of kinetochore–microtubule attachment was less efficient in CENP-C$^{\Delta M12BD}$ RPE-1 cells. As deletion of the M12BD of CENP-C reduced Ndc80C levels, as well as Mis12C and KNL1C, Ndc80C recruitment on CENP-C via Mis12C might be crucial for efficient error correction. However, RPE-1 cells expressing CENP-T lacking the Ndc80C-binding region showed no defects in error correction or chromosome segregation, indicating that Ndc80C reduction is not a major cause of mitotic defects in CENP-C$^{\Delta M12BD}$ RPE-1 cells. We previously demonstrated that kinetochores contain

---

mScarlet-CENP-A was used as a kinetochore marker (CENP-A, red). DNA was stained with DAPI (blue). Scale bar, 10 $\mu$m. The insets show an enlarged single chromosome (scale bar, 1 $\mu$m). H2AT120ph signal intensities at kinetochore-proximal centromeres were quantified (mean and SD, two-tailed *t* test, CENP-C$^{WT}$ RPE-1 cells: n = 398 kinetochores from 5 cells; CENP-C$^{\Delta M12BD}$ RPE-1 cells: n = 373 kinetochores from 5 cells). **(E)** H3T3ph localization in CENP-C$^{WT}$ or CENP-C$^{\Delta M12BD}$ RPE-1 cells. H3T3ph was stained with an antibody against H3T3ph. H3T3ph localization at centromeres was examined and quantified as in (D). Scale bar, 10 $\mu$m. The graph displays the mean and SD (two-tailed *t* test, CENP-C$^{WT}$: n = 170 centromeres from 5 cells; CENP-C$^{\Delta M12BD}$ RPE-1 cells: n = 204 centromeres from 5 cells). **(F)** Bub1 localization in CENP-C$^{WT}$ or CENP-C$^{\Delta M12BD}$ RPE-1 cells. Bub1 was stained with an antibody against Bub1 (green). mScarlet-CENP-A was used as a kinetochore marker (CENP-A, red). DNA was stained with DAPI (blue). Scale bar, 5 $\mu$m. Bub1 signal intensities at kinetochores were quantified (mean and SD, two-tailed *t* test, CENP-C$^{WT}$ RPE-1 cells: n = 318 kinetochores from 5 cells [prophase], n = 354 kinetochores from 5 cells [prometaphase], n = 440 kinetochores from 5 cells [metaphase]; CENP-C$^{\Delta M12BD}$ RPE-1 cells: n = 280 kinetochores from 5 cells [prophase], n = 423 kinetochores from 5 cells [prometaphase], n = 429 kinetochores from 5 cells [metaphase]).

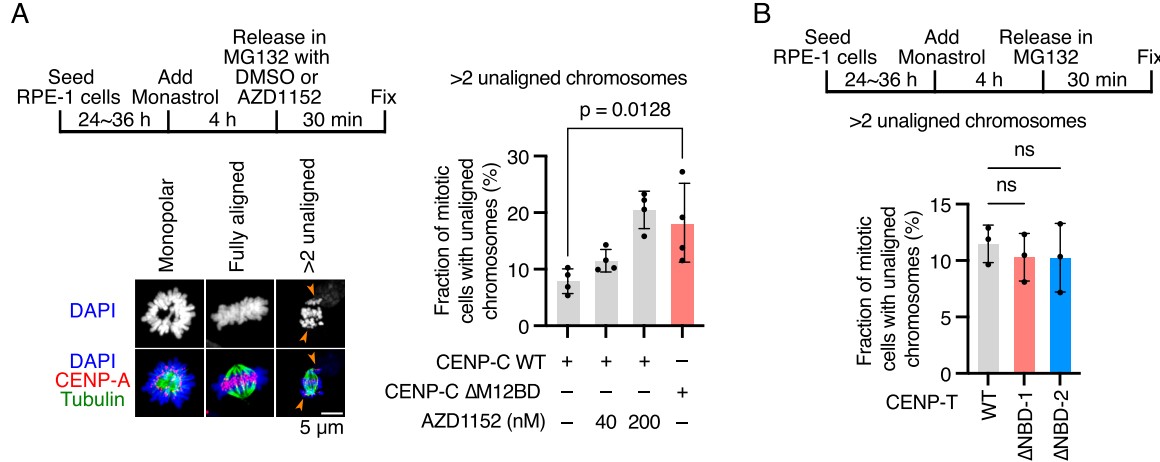

**Figure 6. Deletion of the Mis12C-binding domain of human CENP-C reduces kinetochore–microtubule error correction efficiency in RPE-1 cells.**
**(A)** Error correction assay in CENP-C$^{WT}$ or CENP-C$^{\Delta M12BD}$ RPE-1 cells. The cells were treated with monastrol for 4 h and then released and incubated in a medium with MG132 in the presence or absence of an Aurora B inhibitor (AZD1152 or DMSO) for 30 min. The cells were fixed, and microtubules (green) and DNA (blue) were stained with an antibody against alpha-tubulin and DAPI, respectively. mScarlet-CENP-A was used as a kinetochore marker (CENP-A, red). The cells with more than two unaligned chromosomes were defined as "cells with unaligned chromosomes." Representative images are shown: a monastrol-treated cell (monopolar), a cell with fully aligned chromosomes (fully aligned), and a cell with unaligned chromosomes (>2 unaligned). Arrows indicate unaligned chromosomes. Scale bar, 5 μm. Mitotic cells with unaligned chromosomes were quantified. Four independent experiments were performed (mean and SD, two-tailed t test). **(B)** Error correction assay in CENP-T$^{WT}$, CENP-T$^{\Delta NDB-1}$, or CENP-T$^{\Delta NDB-2}$ RPE-1 cells. Cells were treated with monastrol and released into a medium with MG132, and unaligned chromosomes were quantified as in (A). Three independent experiments were performed (mean and SD, one-way ANOVA with Dunnett's multiple comparison test).

excess Ndc80C molecules via CENP-T (Takenoshita et al, 2022). The remaining Ndc80C levels in the CENP-C$^{\Delta M12BD}$ cells may be sufficient for efficient error correction. Instead, the CENP-C-Mis12C interaction functions through KNL1C, which recruits Bub1. In CENP-C$^{\Delta M12BD}$ RPE-1 cells, Bub1, H2A120ph, and Aurora B levels were reduced in CENP-C$^{\Delta M12BD}$ RPE-1 cells, leading to inefficient error correction of kinetochore–microtubule attachment. Bub1–Aurora B reduction was not observed in the cells expressing CENP-T lacking one of the Ndc80C-binding regions. As Bub1 and BubR1 also play a role in the SAC machinery, the CENP-C-Mis12C interaction may also facilitate the SAC; however, this was not addressed in the current study.

Because CENP-T also recruits Mis12C and KNL1C to kinetochores, the CENP-T-Mis12C interaction may also be involved in the feedback loop that maintains Aurora B (Fig 7F). Although this possibility cannot be ruled out by our current results, we suggest that the Aurora B feedback loop is specifically mediated by the CENP-C-Mis12C interaction. Firstly, the copy number of CENP-C at the ki-netochore was higher than that of CENP-T in human cells (CENP-C 215 versus CENP-T 72) (Suzuki et al, 2015). Secondly, CENP-C is the major scaffold for recruiting Mis12C during prometaphase, during which error correction is performed (Hara et al, 2018). Thirdly, the CENP-C-Mis12C interaction is largely dependent on DSN1 phos-phorylation by Aurora B (Kim & Yu, 2015; Rago et al, 2015; Petrovic et al, 2016). In fact, our data indicated that Mis12C levels were in-creased on CENP-C but not on CENP-T after deletion of the DSN1 basic motif (Fig S8). In addition, the CENP-T-Mis12C interaction is regulated by both CDK1-mediated CENP-T phosphorylation and Aurora B–mediated DSN1 phosphorylation (Walstein et al, 2021). CDK1-mediated CENP-T phosphorylation promotes the CENP-T-Mis12C interaction without DSN1 phosphorylation by Aurora B in vitro (Huis In't Veld et al, 2016; Walstein et al, 2021), suggesting

that the CENP-T-Mis12C interaction may be less sensitive to Aurora B activity in the cells. Further analyses are necessary to clarify whether CENP-C-Mis12C and CENP-T-Mis12C interactions have distinct roles.

A previous study demonstrated that Aurora B is enriched in the inner centromeres of misaligned chromosomes and reduced in aligned chromosomes in RPE-1 cells (Salimian et al, 2011). This study also showed that the levels of Aurora B–mediated DSN1 phos-phorylation are increased at kinetochores on unaligned chromo-somes with Aurora B enrichment. Based on these observations, feedback control from the kinetochores to the inner centromeres has been proposed for Aurora B levels to sense chromosome biorientation (Salimian et al, 2011). In the present study, we clarified the molecular basis for feedback control: at misaligned chromo-somes, Aurora B facilitates DSN1 phosphorylation and the subse-quent CENP-C-Mis12C interaction, which turns on the feedback loop to promote Aurora B localization, leading to the correction of er-roneous microtubule attachment. Once chromosomes align with the bipolar attachment, Aurora B substrates on the kinetochore, including DSN1, are dephosphorylated through spatial separation mechanisms (Tanaka, 2002; Liu et al, 2009), diminishing the CENP-C-Mis12C interaction and suppressing the regulatory loop. This re-duces Aurora B at the aligned chromosomes and leads to further dephosphorylation of its substrates, such as Hec1, which stabilizes the kinetochore–microtubule attachment. The feedback loop control for Aurora B localization is established by the CENP-C-Mis12C interaction, together with the spatial Aurora B separation mechanism, and contributes to efficient error correction for chromosome biorientation in human cells.

In C33A cells, a carcinoma cell line, DSN1, has a mutation at Ser109, which is the Aurora B phosphorylation site that regulates

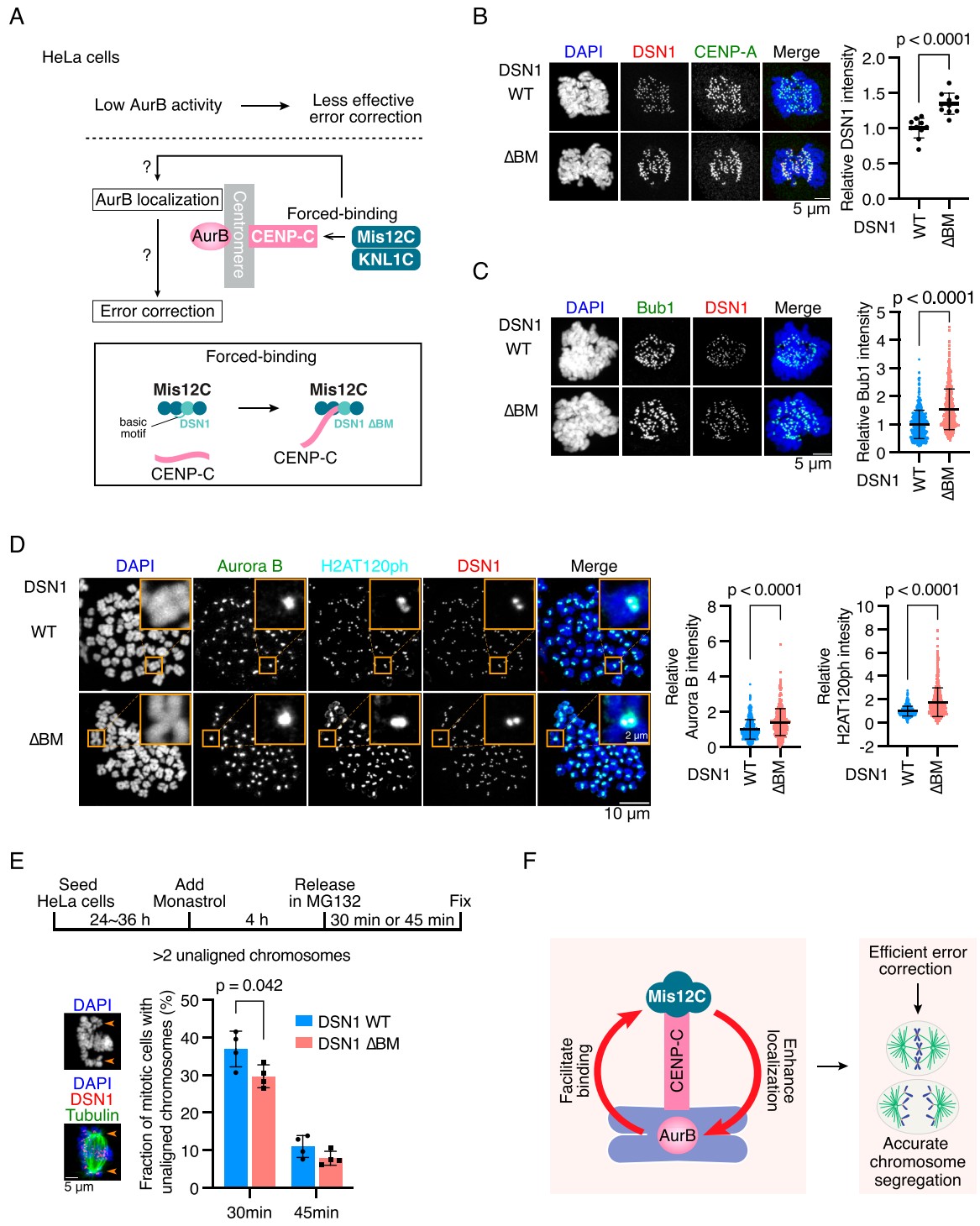

**Figure 7. Forced binding of Mis12C to CENP-C increases Aurora B levels on centromeres and improves error correction in HeLa cells.**
**(A)** Schematic representation for forced binding of Mis12C to CENP-C in HeLa cells. To validate the idea that the CENP-C-Mis12C interaction positively regulates Aurora B localization, we used HeLa cells in which Aurora B activity is low at centromeres, leading to chromosome instability. The CENP-C-Mis12C interaction was increased by expressing a DSN1 mutant lacking the basic motif (ΔBM) in HeLa cells, and we examined the Aurora B levels and efficiency of kinetochore–microtubule error correction. Dsn1 is a subunit of Mis12C. The basic motif of DSN1 masks the CENP-C-binding surface of Mis12C, preventing the CENP-C-Mis12C interaction. The deletion of the basic motif increases the binding affinity between CENP-C and Mis12C (Petrovic et al, 2016). **(B)** DSN1 localization in DSN1$^{WT}$ or DSN1$^{ΔBM}$ HeLa cells. DSN1, a subunit of the Mis12 complex, was stained with an antibody against DSN1 (red). DNA was stained with DAPI (blue). CENP-A was stained by an antibody against CENP-A as a kinetochore marker (green). Scale bar, 5 $\mu$m. DSN1 signal intensities at mitotic kinetochores were quantified (mean and SD, two-tailed $t$ test, DSN1$^{WT}$ HeLa cells: n = 10 cells; DSN1$^{ΔBM}$ HeLa cells: n = 10 cells). **(C)** Bub1 localization in DSN1$^{WT}$ or DSN1$^{ΔBM}$ HeLa cells. Bub1 was stained with an antibody against Bub1 (green). DSN1 was stained as a kinetochore marker (red). DNA was stained with DAPI (blue). Scale bar, 5 $\mu$m. Bub1 signal intensities at kinetochores were quantified (mean and SD, two-tailed $t$ test, DSN1$^{WT}$ HeLa cells:

the CENP-C-Mis12C interaction (DepMap: https://depmap.org/portal/), suggesting that the CENP-C-Mis12C–mediated Aurora B regulatory loop may be impaired in some cancer cells. Indeed, the control of Aurora B levels on unaligned chromosomes is halted in HeLa cells (Salimian et al, 2011), which may be responsible for chromosomal instability in HeLa cells. This is further supported by our finding that the enforced binding of Mis12C to CENP-C increased Aurora B protein levels at the centromeres and improved error correction efficiency in HeLa cells. However, its efficiency was lower than that in RPE-1 cells. The maximum activation of Aurora B kinase requires the binding of the heterochromatin protein HP1 at the mitotic centromeres, and this regulatory mechanism is hampered in cancer cells (Abe et al, 2016), suggesting that in addition to a certain amount of Aurora B at the centromeres, HP1-mediated Aurora B activation is also required to control Aurora B activity for efficient error correction.

The mechanisms controlling Aurora B localization through Bub1/H2AT120ph or the CENP-C-Mis12 interaction by Aurora B have been well studied, separately. Our study highlights that these two mechanisms are integrated into a regulatory system to form a feedback loop for bioriented kinetochore–microtubule attachment. This implies that other regulatory mechanisms that function on centromeres/kinetochores would combine to form regulatory systems. A future understanding of how the regulatory mechanisms are integrated will shed light on the dynamic regulatory systems of centromeres/kinetochores to ensure accurate chromosome segregation, providing potential targets for cancer therapy.

# Materials and Methods

### Establishment of mutant mouse lines

We used electroporation to introduce Cas9 protein and sgRNAs into 1-cell embryos (Hashimoto et al, 2016) to delete exons 2–4. Embryos were obtained from superovulated B6C3F1 females crossed to B6C3F1 males. Embryos were aligned in the 1-mm gap of a CUY501G1 electrode (Nepa Gene) filled with 200 ng/μl of freshly prepared Guide-it Recombinant Cas9 (Electroporation-Ready, Cat #632641; Takara Bio) and 100 ng/μl of each sgRNA in Opti-MEM I (Thermo Fisher Scientific). Electroporation was performed at 30 V (3 msec ON + 97 msec OFF) x 7 pluses using a CUY21 EDIT electroporator (BEX). The embryos were then cultured in modified Whitten's medium (DR01032; PHC Japan) overnight at 37°C under 5% $CO_2$. The embryos that reached the two-cell stage were transferred to the oviducts of pseudopregnant females. Cesarean sections were performed when pregnant females did not deliver naturally, and pups were raised with ICR foster mothers. After genotyping, mice with the deletion were crossed with C57BL/6 to F1 heterozygous mice.

Animal care and experiments were conducted in accordance with the Guidelines of Animal Experiment of the National Institutes of Natural Sciences and the Guide for the Care and Use of Laboratory Animals of the Ministry of Education, Culture, Sports, Science, and Technology of Japan. The experiments employed in this study were approved by the Institutional Animal Care and Use Committee of the National Institutes of Natural Sciences and by the Committee on the Ethics of Animal Experiments of Chiba Cancer Center.

### gRNA synthesis

To delete exons 2–4 of the mouse *Cenpc1* (*Cenpc*) gene, we selected gRNA target sequences intron 1 and intron 4 using CRISPick (Doench et al, 2016; Sanson et al, 2018). gRNAs were synthesized as previously described (Hashimoto et al, 2016). We amplified DNA fragments containing T7 promoter, gRNA target sequences (intron 1: CCAACACTATAGCTGACAAG; intron 4: AAACTGATAGAGTACAGTGG), and gRNA scaffold sequences by PCR using primers shown in Table S1 and pX330 (plasmid #42230; Addgene) (Cong et al, 2013) as a template. After purification of the PCR products by ethanol precipitation, gRNAs were synthesized from the PCR products using MEGA-shortscript T7 Transcription Kit (Thermo Fisher Scientific) and purified by phenol–chloroform–isoamyl alcohol extraction and isopropanol precipitation. gRNAs used in this study are shown in Table S1.

### Establishment of MEFs

MEFs were isolated from 14.5-d-old embryos of $Cenpc^{+/+}$, $Cenpc^{+/ΔM12BD}$, and $Cenpc^{ΔM12BD/ΔM12BD}$ mice. After the removal of the head and organs, the embryos were rinsed with PBS, minced, and subjected to digestion with trypsin–EDTA (0.05%) (Gibco) for 30 min at 37°C. Trypsin was quenched by adding DMEM with high glucose (Sigma-Aldrich) supplemented with 15% FBS. Each digested embryo was then plated on a 100-mm-diameter dish and incubated in humidified air containing 5% $CO_2$ at 37°C. The passage number was documented for each batch of MEFs, and the first passage cells (P1) were cryopreserved in freeze cryopreservation medium (BAM-BANKER, GC LYMPHOTEC) at −80°C.

---

n = 523 kinetochores from 6 cells; DSN1[ΔBM] HeLa cells: n = 501 kinetochores from 6 cells). **(D)** Aurora B and H2AT120ph localization in DSN1[WT] or DSN1[ΔBM] HeLa cells. Aurora B and H2AT120ph were stained with their antibodies (green and cyan). mScarlet-DSN1 was used as a kinetochore marker (DSN1, red). DNA was stained with DAPI (blue). Scale bar, 10 μm. The insets show an enlarged single chromosome (scale bar, 2 μm). The signal intensities of Aurora B at centromeres and H2AT120ph at kinetochore-proximal centromeres were quantified (mean and SD, two-tailed t test, DSN1[WT] HeLa cells: n = 385 centromeres, 651 kinetochores from 6 cells for Aurora B and H2AT120ph, respectively; DSN1[ΔBM] HeLa cells: n = 379 centromeres, 668 kinetochores from 6 cells for Aurora B and H2AT120ph, respectively). **(E)** Error correction assay in DSN1[WT] or DSN1[ΔBM] HeLa cells. The cells were treated with monastrol for 4 h and then released and incubated in a medium with MG132 for 30 or 45 min. The cells were fixed, and microtubules (green) and DNA (blue) were stained with an antibody against alpha-tubulin and DAPI, respectively. mScarlet-DSN1 was used as a kinetochore marker (DSN1, red). The cells with more than two unaligned chromosomes were defined as "cells with unaligned chromosomes." Arrows indicate unaligned chromosomes. Scale bar, 5 μm. Mitotic cells with unaligned chromosomes were quantified. Four independent experiments were performed (mean and SD, two-tailed t test). **(F)** Model for a positive regulatory loop to facilitate Aurora B localization at the centromere through the CENP-C-Mis12C interaction. The regulatory system is required for efficient error correction of kinetochore–microtubule attachment, leading to accurate chromosome segregation.

### Two-step skin carcinogenesis

Sixteen $Cenpc^{+/+}$, 22 $Cenpc^{+/\Delta M12BD}$, and 15 $Cenpc^{\Delta M12BD/\Delta M12BD}$ mice were subjected to the following protocol. At 8–10 wk of age, the backs of the mice were shaved with an electric clipper. 2 d later, DMBA (Sigma-Aldrich) (25 µg per mouse in 200 µl acetone) was applied to the shaved dorsal back skin. 3 d after the first DMBA treatment, 12-O-tetradecanoylphorbol-13-acetate (TAP) (Calbiochem) (10 µg per mouse in 200 µl acetone) was administered. After four rounds of DMBA/TPA treatment, the mice were further treated with TPA twice a week for 20 wk. The number of papillomas was recorded from 8 to 20 wk, and the development of squamous cell carcinoma was monitored for up to 36 wk after TPA treatment.

### Cell culture

Human hTERT-RPE-1 and HeLa Kyoto cell lines were maintained in a culture medium containing DMEM (Nacalai Tesque) supplemented with 10% FBS (Sigma-Aldrich) and penicillin–streptomycin (100 µg/ml) (Thermo Fisher Scientific) and cultured at 37°C, 5% $CO_2$. For degradation of GFP-mAID-CENP-T (RPE-1 cKO-CENP-T), cells were treated with 500 µM of indole-3-acetic acid (IAA; Wako).

For the analysis of MEFs, the P1 MEFs were thawed and subcultured in a culture medium containing DMEM (Nacalai Tesque) supplemented with 10% FBS (Sigma-Aldrich) and penicillin–streptomycin (100 µg/ml) (Thermo Fisher Scientific) at 37°C, 5% $CO_2$. They were then passed one more time (P2) and used for all downstream experiments.

### Plasmid constructions for cell transfection

To express mScarlet-fused full-length CENP-A under the control of the CENP-A promoter in RPE-1 cells, the sequence of *mScarlet-fused CENP-A* followed by the puromycin resistance gene (*PuroR*) or neomycin resistance gene (*NeoR*) expression cassette driven by the *beta-actin* (*ACTB*) promoter was cloned into the pBluescript II SK (pBSK) with 5′ and 3′ homology arm fragments (~1 kb each) surrounding the *CENP-A* start codon (pBSK_mScarlet-CENP-A) using In-Fusion Snap Assembly Master Mix (Takara Bio). To integrate the construct into the endogenous *CENP-A* locus, CRISPR/Cas9-mediated homologous recombination was used, employing pX330 (plasmid #42230; Addgene) (Cong et al, 2013) containing single-guide RNA (sgRNA) targeting a genomic sequence (GGGCCTCGGGCTTTCGGCTC) around the CENP-A start codon (pX330_sgCENP-A). sgRNA for CENP-A was designed using CRISPOR (Concordet & Haeussler, 2018).

To express GFP-fused full-length CENP-A under the control of CMV promoter in RPE-1 cells, the sequence of *GFP-fused CENP-A* followed by the L-histidinol resistance gene (*HisD*) expression cassette driven by the *ACTB* promoter was cloned into the pT2/HB (a gift from Perry Hackett, plasmid #26557; Addgene) (pT2/HB_GFP-CENP-A). The *GFP-fused CENP-A* with the *HisD* expression cassette was integrated into the genome using the Sleeping Beauty transposon system (Mates et al, 2009).

To express GFP-fused histone H2A from the *AAVS1* locus in RPE-1 cells, the sequence of *GFP-fused H2A* followed by the blasticidin S resistance gene (*BsR*) expression cassette driven by the *ACTB* promoter was cloned into the pBSK with 5′ and 3′ homology arm fragments of the *AAVS1* locus (~1 kb each) (pBSK_GFP-H2A) using In-Fusion Snap Assembly Master Mix (Takara Bio). The construct was integrated into the endogenous *AAVS1* locus by CRISPR/Cas9-mediated homologous recombination, employing pX330 (plasmid #42230; Addgene) (Cong et al, 2013) containing sgRNA targeting a genomic sequence (ACCCCACAGTGGGGCCACTA) within intron 1 of *PPP1R112C* (pX330_sgAAVS1). sgRNA for the *AAVS1* locus was designed using CRISPOR (Concordet & Haeussler, 2018).

To express the Flag-tagged full-length human CENP-C (CENP-$C^{WT}$) or ΔMis12C-binding domain (M12BD: aa 1–75) under the control of the endogenous *CENP-C* promoter in RPE-1 cells, the cDNA of *Flag-tagged human CENP-C WT or ΔM12BD* followed by the zeocin (*ZeoR*) or *HisD* expression cassette driven by the *ACTB* promoter was cloned into the pBSK with 5′ and 3′ homology arm fragments (~1 kb each) surrounding the *CENP-C* start codon (pBSK_FLAG-CENP-$C^{WT}$ or FLAG-CENP-$C^{\Delta M12BD}$) using In-Fusion Snap Assembly Master Mix (Takara Bio). The constructs were integrated into the endogenous *CENP-C* locus by CRISPR/Cas9-mediated homologous recombination, employing pX330 (plasmid #42230; Addgene) (Cong et al, 2013) containing sgRNA targeting a genomic sequence (GGCCGGAA-CATGGCTGCGTC) around the *CENP-C* start codon (pX330_sgCENP-C). sgRNA for *CENP-C* was designed using CRISPOR (Concordet & Haeussler, 2018).

To express OsTIR1-T2A-BsR– and GFP-mAID–fused human CENP-T simultaneously under the control of the CMV promoter, the *CENP-T* cDNA was cloned into the pAID1.2-NEGFP (Nishimura et al, 2009; Nishimura & Fukagawa, 2017; Nishimura et al, 2020) (pAID1.2-CMV-NGFP-CENP-T, which includes CMV promoter-OsTIR1-T2A-BsR-IRES2-GFP-mAID-CENP-T). To express OsTIR1-T2A-BsR– and GFP-mAID–fused CENP-T simultaneously under the control of the CMV promoter from the *AAVS1* locus in RPE-1 cells, the sequence of CMV promoter-OsTIR1-T2A-BsR-IRES2-GFP-mAID-CENP-T was cloned into the pBSK with 5′ and 3′ homology arm fragments of the *AAVS1* locus (~1 kb each) (pBSK_AAVS1_OsTIR1_GFP-mAID-CENP-T). The construct was integrated into the endogenous *AAVS1* locus in RPE-1 cells by CRISPR/Cas9-mediated homologous recombination using pX330_sgAAVS1.

Mutant *CENP-T* cDNAs (ΔNBD-1 [Ndc80C-binding domain 1: aa 6–31], ΔNBD-2 [Ndc80C-binding domain 2: aa 76–105, CENP-$T^{\Delta NBD-1}$ and CENP-$T^{\Delta NBD-2}$, respectively] and ΔM12BD [Mis12C-binding domain: aa 107–230, CENP-$T^{\Delta M12BD}$]) were generated using PCR and In-Fusion Snap Assembly Master Mix (Takara Bio). To express the mScarlet-fused full-length CENP-T (CENP-$T^{WT}$), CENP-$T^{\Delta NBD-1}$, CENP-$T^{\Delta NBD-2}$, or CENP-$T^{\Delta M12BD}$ under the control of the endogenous *CENP-T* promoter in RPE-1 cells, the cDNA of *mScarlet-fused CENP-T WT, ΔNBD-1, ΔNBD-2, or ΔM12BD* followed by the *NeoR* or *PuroR* expression cassette driven by the *ACTB* promoter was cloned into the pBSK with 5′ and 3′ homology arm fragments (~1 kb each) surrounding the *CENP-T* start codon (pBSK_mScarlet-CENP-$T^{WT}$, CENP-$T^{\Delta NBD-1}$, CENP-$T^{\Delta NBD-2}$, or CENP-$T^{\Delta M12BD}$). Each construct was integrated into the endogenous *CENP-T* locus in RPE-1 cells by CRISPR/Cas9-mediated homologous recombination, employing pX330 (plasmid #42230; Addgene) (Cong et al, 2013) containing sgRNA targeting a genomic sequence (AGACGATGGCTGACCACAAC) around the *CENP-T* start codon (pX330_sgCENP-T). sgRNA for *CENP-T* was designed using CRISPOR (Concordet & Haeussler, 2018).

To express the mScarlet-fused full-length DSN1 (DSN1^WT) or ΔBM (basic motif, aa 91–113) mutant under the control of the endogenous *DSN1* promoter in RPE1 mScarlet-CENP-A CENP-C^WT or CENP-T^ΔM12BD cells, the cDNA of *mScarlet-fused DSN1 WT* or *ΔBM* followed by the *BsR* expression cassette driven by the *ACTB* promoter was cloned into the pBSK with 5′ and 3′ homology arm fragments (~1 kb each) surrounding the *DSN1* start codon (pBSK_mScarlet-DSN1^WT_BsR or DSN1^ΔBM_BsR) using In-Fusion Snap Assembly Master Mix (Takara Bio). Each construct was integrated into the endogenous *DSN1* locus by CRISPR/Cas9-mediated homologous recombination, employing pX330 (plasmid #42230; Addgene) containing sgRNA targeting a genomic sequence (CTTACCTTGGGTT-CAGGCTT) around the *DSN1* start codon (pX330_sgDSN1). sgRNA for DSN1 was designed using CRISPOR (Concordet & Haeussler, 2018).

To express the mScarlet-fused full-length DSN1 (DSN1^WT) or ΔBasic motif (ΔBM: aa 91–113) mutant under the control of the endogenous *DSN1* promoter in HeLa cells, the cDNA of *mScarlet-fused DSN1 WT* or *ΔBM* followed by the *NeoR* and *PuroR* expression cassette driven by the *ACTB* promoter was cloned into the pBSK with 5′ and 3′ homology arm fragments (~1 kb each) surrounding the *DSN1* start codon (pBSK_mScarlet-DSN1^WT or DSN1^ΔBM) using In-Fusion Snap Assembly Master Mix (Takara Bio). Each construct was integrated into the endogenous *DSN1* locus in HeLa cells by CRISPR/Cas9-mediated homologous recombination, employing pX330 (plasmid #42230; Addgene) containing sgRNA targeting a genomic sequence (CTTACCTTGGGTTCAGGCTT) around the *DSN1* start codon (pX330_sgDSN1). sgRNA for DSN1 was designed using CRISPOR (Concordet & Haeussler, 2018).

To express mouse CENP-C protein (aa 1–405) in *E. coli, mouse CENP-C* (aa 1–405) cDNA was cloned into pET30b (Merck) or pGEX6p-1 (Cytiva).

### Generation of cell lines

To establish RPE-1 cell lines expressing mScarlet-fused CENP-A under the control of the endogenous *CENP-A* promoter, the RPE-1 cells were co-transfected with pBSK_mScarlet-CENP-A and pX330_sgCENP-A using Neon Transfection System (Thermo Fisher Scientific) with 6 pulses (1,400 V, 5 msec), as previously described (Takenoshita et al, 2022). Transfected cells were then subjected to selection in a medium containing 2 mg/ml puromycin (Takara Bio) and 500 µg/ml G418 (Sigma-Aldrich) to isolate single-cell clones (RPE-1 mScarlet-CENP-A cells).

To generate RPE-1 mScarlet-CENP-A cells expressing GFP-fused H2A from the *AAVS1* locus under the control of the CMV promoter, RPE-1 mScarlet-CENP-A cells were co-transfected with pBSK_GFP-H2A and pX330_sgAAVS using Neon Transfection System (Thermo Fisher Scientific) with 6 pulses (1,400 V, 5 msec). Transfected cells were selected in a medium containing 1 mg/ml blasticidin S hydrochloride (Kaken Pharmaceutical) to isolate single-cell clones (RPE-1 mScarlet-CENP-A GFP-H2A cells).

To establish RPE-1 mScarlet-CENP-A or mScarlet-CENP-A GFP-H2A cells expressing either CENP-C WT or ΔM12BD under the control of the endogenous *CENP-C* promoter, mScarlet-CENP-A or mScarlet-CENP-A GFP-H2A RPE-1 cells were co-transfected with pBSK_FLAG-CENP-C^WT or CENP-C^ΔM12BD and pX330_sgCENP-C using

Neon Transfection System (Thermo Fisher Scientific) with 6 pulses (1,400 V, 5 msec). Transfected cells were selected in a medium containing 10 ng/ml zeocin (Invitrogen) and 1.5 mg/ml L-histidinol dihydrochloride (Sigma-Aldrich) to isolate single-cell clones (mScarlet-CENP-A Flag-CENP-C^WT or CENP-C^ΔM12BD RPE-1 cells; mScarlet-CENP-A GFP-H2A Flag-CENP-C^WT or CENP-C^ΔM12BD RPE-1 cells).

To generate an inducible CENP-T protein degradation system using the AID system in RPE-1 cells, the CENP-T AID system expression cassette (CMV promoter-OsTIR1-T2A-BsR-IRES2-GFP-mAID-CENP-T) was integrated into the *AAVS1* locus. RPE-1 cells were co-transfected with pBSK_AAVS1_OsTIR1_GFP-mAID-CENP-T and pX330_sgAAVS1 using Neon Transfection System (Thermo Fisher Scientific) with six pulses (1,400 V, 5 msec). Transfected cells were selected in a medium containing 1 mg/ml blasticidin S hydrochloride (Kaken Pharmaceutical) to isolate single-cell clones (cKO-CENP-T RPE-1 cells).

To generate RPE-1 cKO-CENP-T cells (GFP-mAID-CENP-T) expressing either CENP-T WT, ΔNBD-1, ΔNBD-2, or ΔM12BD, cKO-CENP-T RPE-1 cells were co-transfected with pBSK_mScarlet-CENP-T^WT, CENP-T^ΔNBD-1, CENP-T^ΔNBD-2, or CENP-T^ΔM12BD, and pX330_sgCENP-T using Neon Transfection System (Thermo Fisher Scientific) with six pulses (1,400 V, 5 msec). Transfected cells were selected in a medium containing 2 mg/ml puromycin (Takara Bio) and 500 µg/ml G418 (Sigma-Aldrich) to isolate single-cell clones (cKO-CENP-T mScarlet-CENP-T^WT, CENP-T^ΔNDB-1, CENP-T^ΔNDB-2, or CENP-T^ΔM12BD RPE-1 cells). To express GFP-CENP-A in cKO-CENP-T mScarlet-CENP-T^WT, CENP-T^ΔNBD-1, or CENP-T^ΔNBD-2 RPE-1 cells, we used the Sleeping Beauty transposon system (Mates et al, 2009). The cells were transfected with pT2/HB_GFP-CENP-A and pCMV(CAT)T7-SB100 (plasmid #34879; Addgene) (Mates et al, 2009) selected in a medium containing 1.5 mg/ml L-histidinol dihydrochloride (Sigma-Aldrich) to isolate single-cell clones (cKO-CENP-T GFP-CENP-A mScarlet-CENP-T^WT, CENP-T^ΔNBD-1, or CENP-T^ΔNBD-2 RPE-1 cells).

To establish mScarlet-CENP-A/CENP-C^WT or CENP-C^ΔM12BD RPE-1 cells expressing mScarlet-fused DSN1 WT or ΔBM under the control of the endogenous *DSN1* promoter, RPE-1 cells were co-transfected with pBSK_mScarlet-DSN1^WT_BsR or DSN1^ΔBM_BsR, and pX330_sgDSN1 using Neon Transfection System (Thermo Fisher Scientific) with six pulses (1,400 V, 5 msec). Transfected cells were selected in a medium containing 1 mg/ml blasticidin S hydrochloride (Kaken Pharmaceutical) to isolate single-cell clones (mScarlet-CENP-A/CENP-C^WT/DSN1^WT [CC^WT/DSN1^WT], mScarlet-CENP-A/CENP-C^WT/DSN1^ΔBM [CC^WT/DSN1^ΔBM], mScarlet-CENP-A/CENP-C^ΔM12BD/DSN1^WT [CC^ΔM12BD/DSN1^ΔWT], or mScarlet-CENP-A/CENP-C^ΔM12BD/DSN1^ΔBM [CC^ΔM12BD/DSN1^ΔBM] cells).

To establish HeLa cells expressing mScarlet-fused DSN1 WT or ΔBM under the control of the endogenous *DSN1* promoter, HeLa cells were co-transfected with pBSK_mScarlet-DSN1^WT or DSN1^ΔBM, and pX330_sgDSN1 using Neon Transfection System (Thermo Fisher Scientific) with six pulses (1,400 V, 5 msec). Transfected cells were selected in a medium containing 2 mg/ml puromycin (Takara Bio) and 2 mg/ml G418 (Sigma-Aldrich) to isolate single-cell clones (mScarlet-DSN1^WT or DSN1^ΔBM HeLa cells).

### Cell counting

To count the number of RPE-1 cells or MEFs, the culture medium was aspirated, and then, the cells were washed with PBS. Subsequently,

2.5 g/liter of trypsin and 1 mmol/liter of EDTA solution (Nacalai Tesque) were added and incubated for 3–5 min at RT. Trypsin was quenched by adding the culture medium. The cells were suspended by pipetting several times and then mixed with an equal volume of 0.4 wt/vol% trypan blue solution (Wako). The living cells were counted using Countess II (Thermo Fisher Scientific).

### Genotyping

To extract genomic DNA from MEFs, RPE-1 cells, or HeLa cells, the cells were collected after trypsinization and washed with PBS as described previously (Takenoshita et al, 2022). The collected cells were resuspended in 0.2 mg/ml proteinase K (Sigma-Aldrich) in PBST (0.1% Tween-20 [Nacalai Tesque] in PBS) and then incubated for 90 min at 55°C followed by heating for 15 min at 96°C. The integration of target constructs was confirmed by PCR using primers listed in Table S1.

### Anti-mouse CENP-C antibody generation

Mouse CENP-C aa 1–405 fused with 6 × His were expressed in *E. coli* Rosetta2(DE3) transformed with pET30b-mouse CENP-C$^{1-405}$ and affinity-purified. The purified protein was injected into rabbits to raise antisera (Wako). For affinity purification of mouse CENP-C antibody, GST-fused mouse CENP-C aa 1–405 were expressed in *E. coli* Rosetta2(DE3) transformed with pGEX6P-1-mouse CENP-C$^{1-405}$ and affinity-purified. GST-CENP-C$^{1-405}$ was conjugated with CNBr-Sepharose 4B (Cytiva) and incubated with the antiserum for 1 h at RT. After a wash with 50 mM Tris–HCl, pH 7.5, and 150 mM NaCl, the antibodies were eluted by 200 mM glycine–HCl, pH 2.0, and 150 mM NaCl and immediately neutralized with 1/20 vol. of 1 M Tris. The affinity-purified antibodies were concentrated and then buffer-exchanged to 50 mM Tris–HCl, pH 7.5, and 150 mM NaCl with Amicon Ultra-0.5 ml (Merck).

### Immunoblotting

RPE-1 cells, MEFs, or HeLa cells were collected after trypsinization, washed with PBS, and suspended in 1x Laemmli sample buffer (LSB: 62.5 mM Tris [Trizma base; Sigma-Aldrich]–HCl, pH 6.8, 2% SDS [Nacalai Tesque], 10% glycerol [Nacalai Tesque], 50 mM DTT [Nacalai Tesque], and bromophenol blue [Wako]) to a final concentration of $1 \times 10^4$ cells/$\mu$l. The lysate was sonicated and heated for 5 min at 96°C. Subsequently, the lysate was separated by 5–20% SDS–PAGE (SuperSepAce, Wako) and transferred onto a PVDF membrane (Immobilon-P, Merck). After washing in TBST (0.1% Tween-20 in TBS [50 mM Tris–HCl, pH 7.5, and 150 mM NaCl]) for 15 min, the membrane was incubated with primary antibodies overnight at 4°C. After a 15-min wash with TBST, the membrane was incubated with secondary antibodies for 1 h at RT. After another 15-min wash with TBST, the membrane was incubated with ECL Prime Western Blotting Detection Reagent (Cytiva) for 5 min. The signal was detected and visualized using a ChemiDoc Touch imaging system (Bio-Rad), and the image was processed using ImageLab (Bio-Rad) and Photoshop 2019 (Adobe).

To detect CENP-C signals in RPE-1 CENP-C$^{WT}$ or CENP-C$^{\Delta M12BD}$ cell lines, harvested cells were suspended in TMS buffer (20 mM Tris–HCl, pH 8.0, 5 mM MgCl$_2$, 250 mM sucrose, 0.5% NP-40, and 10% glycerol) for 10 min on ice. The cells were spun down at 17,000*g* for 15 min at 4°C. The pellet was collected and washed with TKS buffer (20 mM Tris–HCl, pH 8.0, 200 mM KCl, 250 mM sucrose, 1% Triton X-100, 10% glycerol, and 1 mM DTT). The precipitate was suspended in lysis buffer (50 mM NH$_2$PO$_4$, 50 mM Na$_2$HPO$_4$, 0.3 M NaCl, 0.1% NP-40, and 1 mM DTT). After sonication, the lysate was diluted in 2 × LSB and heated for 5 min at 96°C. The proteins were detected as above.

The primary antibodies used in this study were rabbit anti-mouse CENP-C (1:5,000), guinea pig anti-human CENP-C (1:10,000) (Ando et al, 2002), mouse anti-human CENP-A (1:3,000) (Ando et al, 2002), rabbit anti-human DSN1 (1:5,000) (a gift from Iain Cheeseman, Whitehead Institute, MIT) (Kline et al, 2006), rat anti-RFP (1:1,000) (ChromoTek), rat anti-human CENP-T (1:1,000) (a gift from Kinya Yoda, Nagoya University, Nagoya, Japan), rat anti-histone H3 (1:5,000) (a gift from Hiroshi Kimura, Tokyo Tech, Tokyo, Japan) (Nozawa et al, 2010), and mouse anti-α-tubulin (1:10,000) (Sigma-Aldrich). The secondary antibodies were HRP-conjugated anti-rabbit IgG (1:10,000) (Jackson ImmunoResearch), HRP-conjugated anti-guinea pig IgG (1:10,000) (Sigma-Aldrich), HRP-conjugated anti-mouse IgG (1:10,000) (Jackson ImmunoResearch), and HRP-conjugated anti-rat IgG (1:10,000) (Jackson ImmunoResearch). All antibodies were diluted in Signal Enhancer Hikari (Nacalai Tesque) to enhance signal sensitivity and specificity.

### Immunofluorescence staining and image acquisition

For the analysis of DSN1, KNL1, CENP-A, Bub1, BubR1, and Ska3, RPE-1 or HeLa cells were seeded onto 35-mm glass-bottom culture dishes (MatTek) for 24–36 h. For the analysis of PLK1, RPE-1 cells were seeded onto 35-mm glass-bottom culture dishes (MatTek) overnight to allow for proper attachment. The cells were synchronized using the double thymidine block protocol. Briefly, cells were treated with 2 mM thymidine for 24 h, followed by a release in fresh medium for 9 h, and then a second thymidine treatment for an additional 18 h. After synchronization, cells were released for 6 h and then treated with 1 $\mu$g/ml nocodazole for 4 h (Fig S5C). The samples were fixed with 4% PFA (Electron Microscopy Sciences) in PHEM buffer (60 mM Pipes, 25 mM Hepes, 10 mM EGTA, and 2 mM MgCl2, pH 6.8) for 10 min at RT. The cells were permeabilized with 0.5% Triton X-100 in PBS for 10 min at RT, followed by blocking with antibody dilution buffer (AbDil, 3% BSA, 0.1% Triton X-100, and 0.1% Na-azide in TBS) for 5–10 min at RT. The samples were then incubated with primary antibodies for 1 h at 37°C or overnight at 4°C. After washing the samples with PBST (0.1% Triton X-100 in PBS) three times for 5 min each, they were incubated with secondary antibodies for 1 h at RT. Subsequently, the samples were washed in PBST three times for 5 min each, followed by staining DNA in 0.1 $\mu$g/ml DAPI (Roche) in PBS for 10 min at RT. After washing with PBS once, cells were mounted with VECTASHIELD Mounting Medium (Vector Laboratories).

For the analysis of K-fiber, RPE-1 cells were seeded onto 35-mm glass-bottom culture dishes (MatTek) for 24–36 h, and supplemented with MG132 (50 $\mu$M) for 1 h. After rinsing the cells with culture medium, the cells were incubated with CaCl$_2$ buffer (1 mM MgCl$_2$, 1 mM CaCl$_2$, 0.5% Triton X-100, and 100 mM Pipes, adjusted to pH 6.8) for 1 min at 37°C. The cells were then fixed in 1%

glutaraldehyde (Nacalai Tesque) in PHEM for 10 min at 37°C. To quench the reaction, 0.1 g/ml sodium tetrahydridoborate (Nacalai Tesque) in PHEM was added and incubated for 20 min at RT. The cells were permeabilized with 0.5% Triton X-100 in PBS for 10 min and blocked with AbDil for 5–10 min at RT. The cells were incubated with primary antibody for 1 h at 37°C, followed by the aforementioned protocol for secondary antibody staining and counterstaining with DAPI.

For α-tubulin staining in the error correction assay, RPE-1 or HeLa cells were seeded onto 35-mm glass-bottom culture dishes (MatTek) for 24–36 h. The cells were then fixed and permeabilized with 3.2% PFA, 0.5% Triton X-100, and 1% glutaraldehyde in PHEM for 10 min at RT, followed by blocking with AbDil for 5–10 min at RT. Subsequently, the cells were incubated with primary antibodies for 1 h at 37°C. The secondary antibody staining and counterstaining with DAPI were performed as above.

For micronucleus analysis, MEFs or RPE-1 cells were seeded onto 35-mm glass-bottom culture dishes (MatTek) and cultured for 2 d. The cells were then fixed and permeabilized with a solution containing 4% PFA and 0.5% Triton X-100 in PHEM buffer for 10 min at RT. Nuclei were stained with DAPI as above.

For the analysis of Aurora B, H3T3ph, and H2AT120ph, chromosome spread samples (Figs 5A, B, D, and E, 7D, S6A and B, S7K, and S8G and H) were used. The cells were cultured with 100 ng/ml nocodazole (Sigma-Aldrich) for 2–4 h at 37°C. To detect Aurora B at kinetochore-proximal pool (Fig S8H), nocodazole treatment followed with 5 $\mu$M 5-iodotubercidin (5-ITu; TOCRIS), a Haspin inhibitor, for 30 min. To spread and swell chromosomes, the method was modified from the protocol described previously (Erin et al, 2021 Preprint). After shaking off and collecting the mitotic-arrested cells, they were suspended in a hypotonic buffer (75 mM KCl [Nacalai Tesque]: 0.8% sodium citrate [Nacalai Tesque]: water at 1:1:1) with 1x cOmplete EDTA-free Protease Inhibitor Cocktail (Roche) on ice. The suspended cells were cytospun onto coverslips using CytoSpin III Cytocentrifuge and fixed in 2% PFA in PBS for 20 min. The cells were blocked in 1% BSA in PBS for 10 min at RT, followed by incubation with primary antibody for 1 h at 37°C or overnight at 4°C. Subsequently, the above protocol for secondary antibody staining and counterstaining with DAPI was performed.

Primary antibodies used were diluted in AbDil, except for Aurora B (diluted in 1% BSA/PBS). The primary antibodies included rabbit anti-human Hec1 (1:5,000) (ab3613; Abcam), mouse anti-human CENP-A (1:250) (Ando et al, 2002), mouse anti-human Aurora B (1:500) (611082; BD Bioscience), rabbit anti-human DSN1 (1:2,000) (a gift from Iain Cheeseman, Whitehead Institute, MIT) (Kline et al, 2006), rabbit anti-human KNL1 (1:2,000) (a gift from Iain Cheeseman, Whitehead Institute, MIT) (Cheeseman et al, 2008), mouse anti-human Bub1 (1:400) (MAB3610; Millipore), rabbit anti-H2AT120ph (1:1,000) (Active Motif), mouse anti-BubR1 (1:500) (MAB3612; Millipore), mouse anti-PLK1 (1:500) (ab17057; Abcam), rabbit anti-Ska3 (1:500) (a gift from Gary J Gorbsky) (Daum et al, 2009), mouse anti-H3T3ph (1:3,000) (Kimura et al, 2008), FITC-conjugated mouse anti-α-tubulin (1:1,000) (F2168; Sigma-Aldrich), and mouse anti-α-tubulin (1:5,000) (Sigma-Aldrich). Secondary antibodies used in immunofluorescence staining were FITC-conjugated goat anti-mouse IgG (1:1,000) (Jackson ImmunoResearch), FITC-conjugated goat anti-

rabbit IgG (1:1,000) (Jackson ImmunoResearch), Alexa 647–conjugated goat anti-mouse IgG (1:1,000) (Jackson ImmunoResearch), and Alexa 647–conjugated goat anti-rabbit IgG (1:1,000) (Jackson ImmunoResearch).

Immunofluorescence images were captured with a spinning disk confocal unit CSU-W1 or CSU-W1-SoRa (Yokogawa) controlled with NIS-Elements (v5.42.01; Nikon) with an objective lens (PlanApo VC 60×/1.40 or Lambda 100×/1.45 NA; Nikon) and an Orca-Fusion BT (Hamamatsu Photonics) sCMOS camera. The images were acquired with Z-stacks at intervals of 0.2 or 0.3 $\mu$m. Maximum intensity projections (MIPs) of the Z-stack were generated using Fiji software (Schindelin et al, 2012) for display and analysis. These images were processed using Fiji and Photoshop 2019 (Adobe).

### Live-cell imaging

For analyzing CENP-C$^{WT}$ or CENP-C$^{\Delta M12BD}$ RPE-1 cells, cells plated onto 35-mm glass-bottom dishes (MatTek) were switched to phenol red–free culture medium (phenol red–free DMEM [Nacalai Tesque], supplemented with 20% FBS, 25 mM Hepes, and 2 mM L-glutamine) and sealed with mineral oil (Sigma-Aldrich). Images were captured every 2 min with the DeltaVision Elite imaging system (GE Healthcare) equipped with a PlanApo N OSC 60×/1.40 NA oil immersion objective lens (Olympus) and a CoolSNAP HQ2 CCD camera (Photometrics) controlled with built-in SoftWoRx software (version 5.5) in a temperature-controlled room at 37°C. A Z-series of 7 sections with 2-$\mu$m increments were acquired. The Z-series was then projected using the MIP for analysis by SoftWoRx. For analyzing CENP-T$^{WT}$, CENP-T$^{\Delta NBD-1}$, or CENP-T$^{\Delta NBD-2}$ RPE-1 cells, the cells plated onto 35-mm glass-bottom dishes supplemented with IAA (500 $\mu$M) for 2 d were switched to phenol red–free culture medium supplemented with IAA (500 $\mu$M) and SPY505-DNA (1:1,000) (Spirochrome) for 4–6 h before imaging. For analyzing Cenpc$^{+/+}$, Cenpc$^{+/\Delta M12BD}$, or Cenpc$^{\Delta M12BD/\Delta M12BD}$ MEFs, the cells plated onto 35-mm glass-bottom dishes were switched to phenol red–free culture medium supplemented with SPY650-DNA (1:3,000) (Spirochrome) for 4–6 h before imaging.

Images were captured every 2 min with the CSU-W1-SoRa system described above at 37°C under the 5% CO$_2$ condition. A Z-series of 7 sections were taken at intervals of 2 $\mu$m. Acquired time-lapse images were projected by the MIP and processed using Fiji and Photoshop 2019 (Adobe).

For chromosome oscillation analysis, cells were treated with SiR-tubulin (1:5,000) (Spirochrome) for 4–6 h and then supplemented with MG132 (5 $\mu$M) for 2 h. Images were filmed every 3 s for 5 min, and a Z-series of 5 or 7 sections in 0.5-$\mu$m increments. The deviation from the average position (DAP) was determined according to the previously described method (Stumpff et al, 2008; Iemura & Tanaka, 2015). In brief, the Z-stack images were deconvolved using the plug-in function of NIS-Elements (Richardson–Lucy method) and then projected by the MIP for quantification. Individual kinetochores were tracked by Manual Tracking (plug-in of Fiji; http://rsb.info.nih.gov/ij/plugins/track/track.html) after aligning the cell movement using the StackReg (plug-in of Fiji) (Thevenaz et al, 1998). The obtained data were analyzed in Microsoft Excel.

## Flow cytometry

MEFs were collected after trypsinization. The harvested cells were washed twice with ice-cold 1% BSA/PBS, fixed with ice-cold 70% ethanol, and stored at –20°C. The fixed cells were washed again with 1% BSA/PBS and incubated with 20 µg/ml propidium iodide (Sigma-Aldrich) in 1% BSA/PBS for 20 min at RT, followed by overnight incubation at 4°C. The stained cells were then subjected to flow cytometry analysis using BD FACSCanto II Flow Cytometer and analyzed with BD FACSDiva 9.0 software (BD Biosciences).

## Cell viability assay

RPE-1 cells were seeded onto opaque-walled tissue culture plates with a clear bottom (Corning) at a density of 2,500 cells per well and incubated for 12 h. The cells were treated with nocodazole at indicated concentrations for 3 d, with each concentration tested in triplicate. After treatment, RealTime-Glo reagents (RealTime-Glo MT Cell Viability Assay; Promega) were added to the cells and incubated for 1 h at 37°C. Luminescence was measured using GloMax Discover System Microplate Reader (Promega). $IC_{50}$ values were determined using GraphPad Prism 9.5.1 (GraphPad).

## Error correction assay

RPE-1 or HeLa cells were plated and incubated on 35-mm glass-bottom culture dishes (MatTek) for 24–36 h. Subsequently, the cells were treated with 50 µM monastrol (Selleckchem) for 4 h, followed by washing with culture medium four times. The cells were then supplemented with MG132 (5 µM) (Sigma-Aldrich) for indicated timing and fixed (3.2% PFA, 0.5% Triton X-100, and 1% glutaraldehyde in PHEM at RT for 10 min) immediately. After fixation and permeabilization, staining was performed using antibodies and DAPI as an aforementioned protocol.

## Quantification and statistical analysis

The fluorescence signal intensities of DSN1, KNL1, Hec1, Bub1, and mScarlet-DSN1 on kinetochores were quantified using Imaris (Bitplane). The fluorescence signal intensities of Aurora B and H3T3ph within the inner centromere regions were measured using Fiji. The signal intensities were quantified by subtracting background signals in a 15-pixel circular region of the chromosome adjacent to the inner centromere region from the signals in a 15-pixel circular region of the inner centromere region. The fluorescence signal intensities of H2AT120ph within the kinetochore-proximal centromere region were measured using Fiji. The signal intensities were quantified by subtracting background signals in a chromosomal region from the signals in a 10-pixel circular region circular area centered on the kinetochore. For quantification of the k-fiber signal intensities, to select the spindle area in a cell, thresholding is applied to the tubulin signals (MIP) using Fiji. The mean signal intensities in the selected area are measured. The value is subtracted by the mean background signal intensities adjacent to the spindle area and quantified as the averaged k-fiber

signals. Data processing was carried out using Microsoft Excel and GraphPad Prism 9.5.1 (GraphPad), and *P*-values were calculated using a two-tailed *t* test or a one-way or two-way ANOVA test followed by multiple comparison tests, or Tukey's multiple comparison test. Each experiment was repeated four times (Fig 4F), three times (Figs 3C, D, I, and J, 4B and E, 5A, B, D, and F, and 7B), or two times (Figs 3B, 4C and D, 5E, and 7C and D) when not mentioned in figure legends, and representative data of replicates were presented.

# Supplementary Information

# Acknowledgements

The authors are very grateful to members of the Fukagawa Lab for the fruitful discussion. We also thank R Fukuoka, K Oshimo, Y Kubota, M Nakagawa, A Kanie, N Yasue, and S Oka for their technical assistance. We also thank GJ Gorbsky, IM Cheeseman, H Kimura, R Nozawa, and T Hirota for their reagents. This work was supported by CREST of JST (JPMJCR21E6); JSPS KAKENHI Grant Numbers 20H05389, 21H05752, 22H00408, 22H04692, and 24H02281 to T Fukagawa; and JSPS KAKENHI Grant Numbers 16K18491, 21H02461, 22H04672, and 24K02005, Takeda Science Foundation, and Daiichi Sankyo Foundation of Life Science to M Hara. This work was also supported by the JSPS KAKENHI Grant Number JP22H04926, the Grant-in-Aid for Transformative Research Areas—Platforms for Advanced Technologies and Research Resources, "Advanced Bioimaging Support," and the NIBB Collaborative Research Program (19-352, 20-305, 21-238, and 22NIBB327). The images of the mouse and pipette in Fig 2A are from Togo TV (© 2016 DBCLS TogoTV, CC-BY-4.0).

## Author Contributions

W Kong: data curation, formal analysis, and investigation.
M Hara: conceptualization, data curation, formal analysis, supervision, funding acquisition, investigation, and writing—original draft, review, and editing.
Y Tokunaga: formal analysis and investigation.
K Okumura: formal analysis and investigation.
Y Hirano: formal analysis and investigation.
J Miao: investigation.
Y Takenoshita: resources.
M Hashimoto: resources.
H Sasaki: resources.
T Fujimori: resources.
Y Wakabayashi: resources.
T Fukagawa: conceptualization, supervision, funding acquisition, project administration, and writing—original draft, review, and editing.

## Conflict of Interest Statement

The authors declare that they have no conflict of interest.

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
