## [Reviewer comments · Life Science Alliance]

Life Science Alliance

CENP-C-Mis12 complex establishes a regulatory loop through Aurora B for chromosome segregation

Weixia Kong, Masatoshi Hara, Yurika Tokunaga, Kazuhiro Okumura, Yasuhiro Hirano, Jiahang Miao, Yusuke Takenoshita, Masakazu Hashimoto, Hiroshi Sasaki, Toshihiko Fujimori, Yuichi Wakabayashi, and Tatsuo Fukagawa

DOI: <https://doi.org/10.26508/lsa.202402927>

Corresponding author(s): *Tatsuo Fukagawa, Osaka University and Masatoshi Hara, Osaka University*

Review Timeline:

Submission Date:	2024-07-04
Editorial Decision:	2024-07-08
Revision Received:	2024-10-06
Editorial Decision:	2024-10-07
Revision Received:	2024-10-08
Accepted:	2024-10-08

Transaction Report:

Please note that the manuscript was previously reviewed at another journal and the reports were taken into account in the decision-making process at Life Science Alliance.

Reviewer #1 Review

Comments to the Authors (Required):

In the manuscript from Kong et al, they determined the effects of deleting the Mis12 Complex binding domain (M12BD) of CENP-C in mice and in various human cell lines. In mice, they found that the number of homozygous mutant offspring were lower than expected and this deficiency skewed towards an unexamined female-specific effect. Further, they found that the M12BD mutant accelerated tumor progression in mice treated with carcinogenic compounds. In MEFs, homozygous deletion of the M12BD causes defects in segregation and delays in cell cycle timing. Nonetheless, they find that there are no gross defects in the homozygous mutant mice. Then, switching to near-diploid RPE1 cells, they find that the CENP-C interaction is important for the recruitment of Mis12C and KNL1, as shown previously by multiple labs. They go on to suggest that the main reason for segregation defects in mutant cells is due to decreased levels of Aurora B at the inner centromere, due to lower levels of Bub1 kinase, which is recruited by KNL1. Lastly, they switch to HeLa cells and demonstrate that a mutant Mis12C that promotes constitutive binding to CENP-C increases both Bub1 and Aurora B levels at centromeres/kinetochores, and suggest this improves the poor error correction rates of this cell line. They propose that the CENP-C-Mis12C interaction regulates Aurora B levels at centromeres.

I have no major technical issues with this manuscript, and it is generally clear and well carried out. However, it does not provide a lot of new data: It has been previously shown that CENP-C regulates Mis12C and KNL1 Kt levels, that KNL1 regulates Bub1 Kt levels, and that Bub1 regulates the levels of Aurora B. The advance here would seem to be linking all of these previously known linkages together. However, they cannot attribute all observed defects in segregation and cell cycle timing to misregulation of Aurora B since lowered levels of Mis12C and KNL1 would also lead to defects in BubR1, PP2A, PP1, Polo Kinase and checkpoint proteins, to name a few. Of course, all of these factors are connected, and these networks have already been mapped out and discussed elsewhere. In my opinion, the authors overestimate the novelty of their findings, possibly because of their previous work focused on the importance of CENP-T. A stronger case needs to be made throughout the manuscript why this is an advance to our understanding of the role of CENP-C in kinetochore function.

Major Points

1) The authors try to attribute the observed defects in alignment and segregation in their CENP-C mutants on the mis-regulation of Aurora B. However, as noted above, Mis12C regulates many other important proteins at the kinetochore. They should measure the levels of PP2A, BubR1, Polo, Ska Complex, and/or CENP-E at kinetochores in their CENP-C mutant. Given current knowledge, these factors would likely be perturbed, making it difficult to pin the effects of their mutants solely/mainly on Aurora B.

2) The main strength of this paper is the mouse work, where they show that the CENP-C M12BD mutant does not demonstrate gross defects in the development of the mouse. They use this observation to suggest that CENP-T is therefore the more important factor for kinetochore assembly. However, to support this, they would need to generate a mouse model that interferes with CENP-T's ability to interact with Ndc80C or Mis12C (or both). It is entirely possible that there are multiple redundant mechanisms, and that similarly precise CENP-T mutants would also support the development of generally normal mice.

3) The defects in the overall number of homozygous mutant offspring (less than 25%) and in homozygous females (30% of homozygous) is intriguing. Therefore, there seems to be a disadvantage to being XX. If the authors could provide some mechanistic insight into this, that would greatly strengthen the paper in my view. Are there defects in fertility in crosses of homozygous male

and female mice? Again, this would strengthen the paper and make a stronger case for the importance of CENP-C-Mis12C interaction in the germline and early embryo.

4) There are a lot of comparisons of the CENP-C Mis12C binding mutant to CENP-T mutants lacking one of its direct Ndc80 binding domains in the manuscript, but a more apt comparison would be to a Mis12-binding mutant of CENP-T and/or to an N-terminal deletion of CENP-T.

5) It is unclear why the authors switched from RPE-1 cells to HeLa cells near the end of the Results section. Their results in HeLa cells, particularly the error correction data in Figure 7E, are not particularly convincing. Instead, the authors should see if their Dsn1 mutant can rescue the CENP-C M12BD mutant defects in error correction, segregation, and cell cycle timing in RPE-1 cells. They should check the effects of the Dsn1 mutant alone as well. It should be noted however that this Dsn1 mutant will also increase the binding of Mis12C to CENP-T as well. Bub1 controls a kinetochore-adjacent pool of Aurora B (work from DeLuca, Lens, and Lacefield labs and more). If their model is correct, then the Dsn1 mutant should increase the amount of Aurora B adjacent to kinetochores.

Minor Points

1) In Figure 1C, it is confusing how the normalization of cell number was performed. If the homozygous mutant is starting off with fewer cells, that would give it a disadvantage to growth compared to wild-type. In fact, the linear rates look similar between all of the conditions. Further, the spatial arrangement of MEFs affects their growth, and no images of cells are given. Please explain this data more clearly or take it out.

2) I believe the incorrect citation is given for the control of chromosome oscillations by Aurora B on page 10, last paragraph (Zaytsev et al). Instead, PMID: 28552353 and PMID: 21266467 are more appropriate references, wherein they show that changes to Hec1 phosphorylation affect oscillations.

Reviewer #2 Review

Comments to the Authors (Required):

Accurate chromosome segregation relies on the ability of cells to ensure timely kinetochore assembly and establishment of correct kinetochore-microtubule attachments. To achieve this, cells employ multiple regulatory mechanisms involving tightly coordinated kinase-phosphatase activities. Here, Kong et al., report a novel regulatory loop between the kinetochore protein CENP-C and a major mitotic regulatory kinase Aurora B (enzymatic core of the Chromosomal Passenger Complex, CPC), implicated in the regulation of sister chromatid cohesion and error-correction. They show that CENP-C interaction with the outer kinetochore Mis12 complex facilitates Aurora B recruitment to centromere. Perturbing this regulatory loop compromises high-fidelity chromosome segregation in human RPE1 cells.

Kong et al., show that perturbing CENP-C-Mis12 complex interactions did not severely affect mouse development, however, resulted in mitotic defects in mouse embryonic fibroblasts. Exploiting, a skin carcinogenesis model they highlight the importance of the CENP-C-Mis12 complex interaction, as deleting the Mis12 binding region of CENP-C accelerated tumour formation. Using a partial Ndc80 binding deficient CENP-T mutant they show that reduced Ndc80 and reduced K-fiber do not cause major mitotic errors. Interestingly, perturbing CENP-C - Mis12 complex interaction reduced the inner centromere localisation of Aurora B. They further demonstrate that H2AT120ph, one of the two

histone phosphorylations (H3T3ph being the other one) is reduced in cells lacking CENP-C - Mis12 complex interaction. They could rescue the defect by forcing CENP-C - Mis12 complex interaction using a DSN1 mutant in HeLa cells.

Overall, it is a solid study. It has been established previously that H2AT120ph contributes to the KT proximal pool of CPC/Aurora B. The conclusions of this work will be strengthened if the authors assess the changes in the inner centromere pools of CPC/Aurora B (inner centromere or H2AT120ph mediated kinetochore proximal) upon perturbing CENP-C - Mis12 complex interaction by quantifying different CPC pools within the centromere using chromosome spreads. The work presented uses different cell lines (MEF, RPE-1 and HeLa) for different questions; it would be helpful if justifications for moving from one cell line to another could be discussed explicitly.

July 8, 2024

Re: Life Science Alliance manuscript #LSA-2024-02927-T

Prof. Tatsuo Fukagawa
Osaka University
FBS
Yamadaoka 1-3
Suita 5650871
Japan

Dear Dr. Fukagawa,

Thank you for submitting your manuscript entitled "CENP-C-Mis12 complex establishes a regulatory loop through Aurora B for chromosome segregation" to Life Science Alliance. We invite you to submit a revised manuscript addressing the following Reviewer comments:

- Address Reviewer 1's Major Points #1, 4 & 5, as well as the Minor Points. Major Point #3 can be addressed with any relevant data already available.
- Address Reviewer 2's comments.

Thank you for this interesting contribution to Life Science Alliance. We are looking forward to receiving your revised manuscript.

Sincerely,

- A letter addressing the reviewers' comments point by point.
- An editable version of the final text (.DOC or .DOCX) is needed for copyediting (no PDFs).
- High-resolution figure, supplementary figure and video files uploaded as individual files: See our detailed guidelines for preparing your production-ready images, <https://www.life-science-alliance.org/authors>
- Summary blurb (enter in submission system): A short text summarizing in a single sentence the study (max. 200 characters including spaces). This text is used in conjunction with the titles of papers, hence should be informative and complementary to the title and running title. It should describe the context and significance of the findings for a general readership; it should be written in the present tense and refer to the work in the third person. Author names should not be mentioned.
- By submitting a revision, you attest that you are aware of our payment policies found here: <https://www.life-science-alliance.org/copyright-license-fee>

B. MANUSCRIPT ORGANIZATION AND FORMATTING:

Reviewer #1

In the manuscript from Kong et al, they determined the effects of deleting the Mis12 Complex binding domain (M12BD) of CENP-C in mice and in various human cell lines. In mice, they found that the number of homozygous mutant offspring were lower than expected and this deficiency skewed towards an unexamined female-specific effect. Further, they found that the M12BD mutant accelerated tumor progression in mice treated with carcinogenic compounds. In MEFs, homozygous deletion of the M12BD causes defects in segregation and delays in cell cycle timing. Nonetheless, they find that there are no gross defects in the homozygous mutant mice. Then, switching to near-diploid RPE1 cells, they find that the CENP-C interaction is important for the recruitment of Mis12C and KNL1, as shown previously by multiple labs. They go on to suggest that the main reason for segregation defects in mutant cells is due to decreased levels of Aurora B at the inner centromere, due to lower levels of Bub1 kinase, which is recruited by KNL1. Lastly, they switch to HeLa cells and demonstrate that a mutant Mis12C that promotes constitutive binding to CENP-C increases both Bub1 and Aurora B levels at centromeres/kinetochores, and suggest this improves the poor error correction rates of this cell line. They propose that the CENP-C-Mis12C interaction regulates Aurora B levels at centromeres.

I have no major technical issues with this manuscript, and it is generally clear and well carried out. However, it does not provide a lot of new data: It has been previously shown that CENP-C regulates Mis12C and KNL1 Kt levels, that KNL1 regulates Bub1 Kt levels, and that Bub1 regulates the levels of Aurora B. The advance here would seem to be linking all of these previously known linkages together. However, they cannot attribute all observed defects in segregation and cell cycle timing to misregulation of Aurora B since lowered levels of Mis12C and KNL1 would also lead to defects in BubR1, PP2A, PPI, Polo Kinase and checkpoint proteins, to name a few. Of course, all of these factors are connected, and these networks have already been mapped out and discussed elsewhere. In my opinion, the authors overestimate the novelty of their findings, possibly because of their previous work focused on the importance of CENP-T. A stronger case needs to be made throughout the manuscript why this is an advance to our understanding of the role of CENP-C in kinetochore function.

Thank you for useful comments. Although we agree with the Reviewer' points, we believe that our data are still important to understand kinetochore functions in various cell types.

1. The authors try to attribute the observed defects in alignment and segregation in their CENP-C mutants on the mis-regulation of Aurora B. However, as noted above, Mis12C regulates many other important proteins at the kinetochore. They should measure the levels of PP2A, BubR1, Polo, Ska Complex, and/or CENP-E at kinetochores in their CENP-C mutant. Given current knowledge, these factors would likely be perturbed, making it difficult to pin the effects of their mutants solely/mainly on Aurora B.

Per the suggestion of this comment, we measured the levels of BubR1, PLK1, and Ska Complex (Ska 3) in RPE-1 cells expressing CENP-C N-terminus mutant (CENP-C^{ΔM12BD} RPE-1 cells). As shown in the Supplementary revised Figure 5, BubR1, PLK1, and Ska 3 levels were significantly reduced.

3. The defects in the overall number of homozygous mutant offspring (less than 25%) and in homozygous females (30% of homozygous) is intriguing. Therefore, there seems to be a disadvantage to being XX. If the authors could provide some mechanistic insight into this, that would greatly strengthen the paper in my view. Are there defects in fertility in crosses of homozygous male and female mice? Again, this would strengthen the paper and make a stronger case for the importance of CENP-C-Mis12C interaction in the germline and early embryo.

A similar female-biased lethality was reported in mouse embryos with mutations that cause defects in DNA replication or replication-associated repair, leading to genomic instability (McNarin et al., 2019). This is because of maleness; testosterone, which has anti-inflammatory activity, protects male embryos against inflammation induced by genome instability (McNarin et al., 2019). Although we speculate that it may also be the case with the *Cenpc*^{ΔM12BD/ΔM12BD} males, which causes chromosome instability, we have not tested it. We added the details mechanisms of female-biased lethality in the previous work on page 6.

Fertility of the *Cenpc*^{ΔM12BD/ΔM12BD} mice is an interesting point; However, we have not examined it rigorously yet. In our preliminary tests, we crossed the homozygous mice with wild-type mice. Both homozygous female and male mice gave rise to viable offspring, suggesting their fertility, yet we cannot rule out the possibility of subfertility.

Parent genotype	Progeny #		
	Female	Male	Total
Cenpc ^{ΔM12BD/ΔM12BD} x Cenpc ^{+/+}	13	13	26
Cenpc ^{+/+} x Cenpc ^{ΔM12BD/ΔM12BD}	14	6	20

4. There are a lot of comparisons of the CENP-C Mis12C binding mutant to CENP-T mutants lacking one of its direct Ndc80 binding domains in the manuscript, but a more apt comparison would be to a Mis12-binding mutant of CENP-T and/or to an N-terminal deletion of CENP-T.

We used CENP-T^{ΔNBD-1} or CENP-T^{ΔNBD-2} RPE-1 cells to rule out the possibility that the reduction of Ndc80C causes mitotic progression defects in CENP-C^{ΔM12BD} RPE-1 cells, because both Mis12C and Ndc80C levels are reduced in the CENP-C^{ΔM12BD} cells. To clarify which complex reduction causes the mitotic defects, we used CENP-T^{ΔNBD-1} or CENP-T^{ΔNBD-2} RPE-1 cells in which Ndc80C was reduced but Mis12C was maintained by deleting one of the Ndc80-binding domains of CENP-T (Figure 4A-D). However, we agree that it is better to look at the phenotype of cells expressing a CENP-T mutant lacking a Mis12C binding. Therefore, we made cell lines expressing the CENP-T^{ΔM12BD} mutant and examined levels of kinetochore proteins and Aurora B. As shown in revised Supplementary Figure 7, Dsn1, KNL1, Hec1, Bub1, H2AT120ph, and Aurora B levels are reduced. The results suggest that the CENP-T-bound Mis12C may also contribute to efficient error correction. However, the feedback from Aurora B at centromeres seems to mainly target CENP-C-binding of Mis12C to form the regulatory loop as we explain below (response to comment 5).

5. It is unclear why the authors switched from RPE-1 cells to HeLa cells near the end of the Results section. Their results in HeLa cells, particularly the error correction data in Figure 7E, are not particularly convincing. Instead, the authors should see if their Dsn1

mutant can rescue the CENP-C M12BD mutant defects in error correction, segregation, and cell cycle timing in RPE-1 cells. They should check the effects of the Dsn1 mutant alone as well. It should be noted however that this Dsn1 mutant will also increase the binding of Mis12C to CENP-T as well. Bub1 controls a kinetochore-adjacent pool of Aurora B (work from DeLuca, Lens, and Lacefield labs and more). If their model is correct, then the Dsn1 mutant should increase the amount of Aurora B adjacent to kinetochores.

Compared with RPE-1 cells, HeLa cells, a cancer cell line, have low Aurora B activity with inefficient error correction. We used HeLa cells as a model to prove the idea that the CENP-C-bound DSN1 positively regulates the Aurora B localization for the efficient error correction. We clarify why we used HeLa cells in the revised version. Expression of the DSN1 basic motif deletion mutant (Δ BM), which appears to increase CENP-C-bound Mis12C (as below), improved error correction efficiency slightly but significantly in HeLa cells (Figure 7). We agree that the change is small, however it was surprising that the simple DSN1 ^{Δ BM} expression ameliorated the error correction in the cancer cells.

In addition, as per the suggestion of the Reviewer, we made CENP-C M12BD mutant RPE-1 cells expressing DSN1 ^{Δ BM}. We also made wild-type RPE-1 cells expressing the DSN1 basic motif mutant. As shown in revised Supplementary Figure 8, DSN1 levels are increased in CENP-C wild-type, but not Δ M12BD, RPE-1 cells upon expression of the Dsn1 basic motif mutant. This indicates that the expression of DSN1 ^{Δ BM} increases Mis12C on CENP-C, but not CENP-T. The result seems to be contrary to expectations based on in vitro observations that Aurora B phosphorylation of DSN1 increases the binding affinity of Mis12C to CENP-T (Walstein et al., 2021) as well as to CENP-C. Our explanation is that almost all CENP-T molecules at kinetochores are occupied by Mis12C in the cells (Suzuki et al., 2015; CENP-T : Mis12C = 72 : 69 at a kinetochore), but Mis12C-free CENP-C exists at kinetochores (Suzuki et al., 2015; CENP-C : Mis12C = 215 : 82 at a kinetochore). There is no CENP-T vacancy on CENP-T molecules for extra Mis12C even when their affinity is increased by the basic motif deletion from DSN1. In contrast, as there is a large pool of Mis12C-free CENP-C at kinetochores, these CENP-C molecules would accept Mis12Cs and increase their levels at kinetochores, upon expression of DSN1 ^{Δ BM} in the CENP-C wild-type cells. This strongly supports our idea that the feedback from Aurora B at centromeres mainly targets CENP-C-binding of Mis12C to form the regulatory loop.

Concerning a kinetochore-adjacent pool of Aurora B, we examined the Aurora B levels at kinetochore proximal regions using the previously described protocol (Broad et al., 2020). We treated cells with a Haspin inhibitor to emphasize the kinetochore-adjacent and found that the kinetochore-adjacent pool of Aurora B was increased by the expression of DSN1 ^{Δ BM} in CENP-C wild-type RPE-1 cells (Supplementary Figure 8H).

Minor points

1. In Figure 1C, it is confusing how the normalization of cell number was performed. If the homozygous mutant is starting off with fewer cells, that would give it a disadvantage to growth compared to wild-type. In fact, the linear rates look similar between all of the conditions. Further, the spatial arrangement of MEFs affects their growth, and no images of cells are given. Please explain this data more clearly or take it out.

We seeded the same number of cells at day 0. In the previous version, we used a log scale for the y-axis and curve fitting to draw the growth line. These appeared to cause confusion. Now, we revised the growth graph using the

same data set in the revised version (Figure 1C). We also presented cell images on day 2.

2. I believe the incorrect citation is given for the control of chromosome oscillations by Aurora B on page 10, last paragraph (Zaytsev et al). Instead, PMID: 28552353 and PMID: 21266467 are more appropriate references, wherein they show that changes to Hec1 phosphorylation affect oscillations.

Per the suggestion of this Reviewer, we added the suggested citation on page 10 in the revised version.

Reviewer #2

Accurate chromosome segregation relies on the ability of cells to ensure timely kinetochore assembly and establishment of correct kinetochore-microtubule attachments. To achieve this, cells employ multiple regulatory mechanisms involving tightly coordinated kinase-phosphatase activities. Here, Kong et al., report a novel regulatory loop between the kinetochore protein CENP-C and a major mitotic regulatory kinase Aurora B (enzymatic core of the Chromosomal Passenger Complex, CPC), implicated in the regulation of sister chromatid cohesion and error-correction. They show that CENP-C interaction with the outer kinetochore Mis12 complex facilitates Aurora B recruitment to centromere. Perturbing this regulatory loop compromises high-fidelity chromosome segregation in human RPE1 cells.

Kong et al., show that perturbing CENP-C-Mis12 complex interactions did not severely affect mouse development, however, resulted in mitotic defects in mouse embryonic fibroblasts. Exploiting, a skin carcinogenesis model they highlight the importance of the CENP-C-Mis12 complex interaction, as deleting the Mis12 binding region of CENP-C accelerated tumour formation. Using a partial Ndc80 binding deficient CENP-T mutant they show that reduced Ndc80 and reduced K-fiber do not cause major mitotic errors. Interestingly, perturbing CENP-C - Mis12 complex interaction reduced the inner centromere localisation of Aurora B. They further demonstrate that H2AT120ph, one of the two histone phosphorylations (H3T3ph being the other one) is reduced in cells lacking CENP-C - Mis12 complex interaction. They could rescue the defect by forcing CENP-C - Mis12 complex interaction using a DSN1 mutant in HeLa cells.

Overall, it is a solid study. It has been established previously that H2AT120ph contributes to the KT proximal pool of CPC/Aurora B. The conclusions of this work will be strengthened if the authors assess the changes in the inner centromere pools of CPC/Aurora B (inner centromere or H2AT120ph mediated kinetochore proximal) upon perturbing CENP-C - Mis12 complex interaction by quantifying different CPC pools within the centromere using chromosome spreads. The work presented uses different cell lines (MEF, RPE-1 and HeLa) for different questions; it would be helpful if justifications for moving from one cell line to another could be discussed explicitly.

As described in response to Reviewer 1, we used a Haspin inhibitor to see the kinetochore proximal pool of Aurora B. We added new data in the revised version (Supplementary Figure 8H).

We emphasized why we used different cell lines for each experiment in the revised version.

October 7, 2024

RE: Life Science Alliance Manuscript #LSA-2024-02927-TR

Prof. Tatsuo Fukagawa
Osaka University
FBS
Yamadaoka 1-3
Suita 5650871
Japan

Dear Dr. Fukagawa,

Thank you for submitting your revised manuscript entitled "CENP-C-Mis12 complex establishes a regulatory loop through Aurora B for chromosome segregation". We would be happy to publish your paper in Life Science Alliance pending final revisions necessary to meet our formatting guidelines.

- please be sure that the authorship listing and order is correct
- please add your supplementary figure legends to the main manuscript text
- please upload your supplementary figures as single files
- please upload your table files as separate editable doc or excel files
- please upload ORCID ID for secondary corresponding author-they should have received instructions on how to do so

Figure Check:

- please add scale bars to Figure 1C

A. FINAL FILES:

B. MANUSCRIPT ORGANIZATION AND FORMATTING:

Sincerely,

October 8, 2024

RE: Life Science Alliance Manuscript #LSA-2024-02927-TRR

Prof. Tatsuo Fukagawa
Osaka University
FBS
Yamadaoka 1-3
Suita 5650871
Japan

Dear Dr. Fukagawa,

Thank you for submitting your Research Article entitled "CENP-C-Mis12 complex establishes a regulatory loop through Aurora B for chromosome segregation". It is a pleasure to let you know that your manuscript is now accepted for publication in Life Science Alliance. Congratulations on this interesting work.

DISTRIBUTION OF MATERIALS:

Again, congratulations on a very nice paper. I hope you found the review process to be constructive and are pleased with how the manuscript was handled editorially. We look forward to future exciting submissions from your lab.

Sincerely,
